# A conserved molecular switch in Class F receptors regulates receptor activation and pathway selection

Shane C. Wright [1], Paweł Kozielewicz [1], Maria Kowalski-Jahn[1], Julian Petersen[1], Carl-Fredrik Bowin[1], Greg Slodkowicz[2], Maria Marti-Solano [2], David Rodríguez[3], Belma Hot[1], Najeah Okashah[4], Katerina Strakova[1], Jana Valnohova[1], M. Madan Babu [2], Nevin A. Lambert [4], Jens Carlsson[3] & Gunnar Schulte [1]

Class F receptors are considered valuable therapeutic targets due to their role in human disease, but structural changes accompanying receptor activation remain unexplored. Employing population and cancer genomics data, structural analyses, molecular dynamics simulations, resonance energy transfer-based approaches and mutagenesis, we identify a conserved basic amino acid in TM6 in Class F receptors that acts as a molecular switch to mediate receptor activation. Across all tested Class F receptors ($FZD_{4,5,6,7}$, SMO), mutation of the molecular switch confers an increased potency of agonists by stabilizing an active conformation as assessed by engineered mini G proteins as conformational sensors. Disruption of the switch abrogates the functional interaction between FZDs and the phospho-protein Dishevelled, supporting conformational selection as a prerequisite for functional selectivity. Our studies reveal the molecular basis of a common activation mechanism conserved in all Class F receptors, which facilitates assay development and future discovery of Class F receptor-targeting drugs.

---

[1] Section of Receptor Biology & Signaling, Dept. Physiology & Pharmacology, Karolinska Institutet, S17165 Stockholm, Sweden. [2] MRC Laboratory of Molecular Biology, Francis Crick Avenue, Cambridge Biomedical Campus, Cambridge CB2 0QH, United Kingdom. [3] Science for Life Laboratory, Department of Cell and Molecular Biology, Uppsala University, P.O. Box 596SE-751 24, Uppsala, Sweden. [4] Department of Pharmacology and Toxicology, Medical College of Georgia at Augusta University, Augusta, Georgia 30912, USA. These authors contributed equally: Shane C. Wright, Paweł Kozielewicz. Correspondence and requests for materials should be addressed to G.S. (email: gunnar.schulte@ki.se)

The Class F of G protein-coupled receptors (GPCRs) is evolutionarily conserved and consists of ten Frizzled paralogs ($FZD_{1-10}$) and Smoothened (SMO) in humans[1]. While FZDs mediate WNT signaling, SMO mediates Hedgehog signaling. Together, these receptors play key roles in embryonic development, stem cell regulation and tumorigenesis[2,3]. Although Class A GPCRs contain a number of well-characterized motifs that are central to mediating receptor activation and selective interaction with heterotrimeric G proteins, similar motifs in Class F receptors are unknown. In fact, the lack of conserved E/DRY (ionic lock), toggle switch or NPxxY motifs has been described as an argument against the GPCR nature of Class F receptors[4,5].

GPCRs function as allosteric machines sampling a range of conformations spanning from inactive to agonist-bound G protein-coupled states. Active states—of which many can exist—allow receptor activation towards different effectors such as heterotrimeric G proteins, arrestins, or G protein-coupled receptor kinases[6]. Furthermore, Class A GPCRs have been described to act as proto-oncogenes through mutations in the ionic lock that promote a ligand-independent active conformation, resulting in G protein coupling beyond physiological constitutive activity[7,8]. To make sense of the structural rearrangements that result in these overactive receptors, we need to refer to the ternary complex model to relate how the receptor-bound ligand and intracellular transducer affect one another through bidirectional allostery[6,9–11] To date, it is not clear what conformational rearrangements in Class F receptors lead to pathway activation as a consequence of agonist binding, irrespective of the nature of the downstream signaling route (e.g., Dishevelled (DVL)- and heterotrimeric G protein-mediated pathways). Nevertheless, there is emerging evidence that SMO and FZDs interact with their respective ligands and heterotrimeric G proteins to form a functional ternary complex reminiscent of Class A/B GPCRs[12–18]. Receptor state-selective nanobodies and engineered heterotrimeric G proteins, so-called mini G (mG) proteins, have provided valuable, biotechnological tools for probing and stabilizing active Class A/B receptor conformation in living cells and offering exciting possibilities in vitro to better understand Class F receptor activation mechanisms[19–24]. Although individual motifs and residues in FZDs have been identified that mediate interaction with the phosphoprotein DVL[25], how this translates into a pathway-selective, three dimensional DVL-bound receptor conformation is currently unknown.

Here, we use a combination of population and cancer genomics data analysis, analysis of available crystal structures and computational modeling to interrogate the pathophysiological importance to the family-wide conserved residue $R/K^{6.32}$ in Class F receptors. This residue plays a central role in the formation of a ligand-receptor-G protein ternary complex as evidenced by the shift in potency of the agonist in the presence of engineered G protein upon mutation of $R/K^{6.32}$. By comparing wild type and mutant Class F receptors, we provide the proof-of-principle that we can detect the fully active, G protein-coupled Class F receptor conformation in living cells and suggest a molecular switch mechanism based on $R/K^{6.32}$ interaction with TM7. Interestingly, mutation of the molecular switch abrogates the interaction and communication with DVL, despite displaying a higher agonist potency in the mG protein recruitment assay. These findings suggest that FZDs show conformational bias towards different transducer proteins and can guide future drug discovery efforts to screen for pathway-selective drugs targeting active Class F receptors in disease.

## Results

**Genomic data analysis defines $R^{6.32}$ as a mutational hot spot.**
In order to shed light on general activation mechanisms in this class of receptors, we focused on conserved residues with putative biological function. Large scale sequence alignment of over 750 mammalian and non-mammalian FZDs and SMO revealed several positions that are conserved among the human paralogs, in mammals as well as across the animal kingdom (Supplementary Figure 1a, b and Supplementary Data). Given the role of Class F receptors in cancers[26], we investigated the importance of the conserved positions by analyzing which positions are significantly mutated in diverse human cancers. Investigation of the recently published data on 66,402 cancer genomes from the cBioPortal for Cancer Genomics[27] and projection of mutation frequency onto a Class F receptor model revealed the mutational hot spots (Fig. 1a and Supplementary Figures 2a, 3a). We observed that a conserved basic residue—either an arginine (R) or a lysine (K)—at the lower part of TM6 (the residue $R/K^{6.32}$ according to the Ballesteros–Weinstein nomenclature[28]) is significantly mutated in a series of human tumors such as colorectal adenocarcinoma in several Class F members (Supplementary Figure 3b). Focusing on $FZD_6$, it becomes obvious that $R416Q^{6.32}$ is the most prevalent variant associated with cancer in Class F receptors. In other FZD paralogs or SMO, mutation of $R^{6.32}$ to H, C, Q, and S is associated with different forms of cancer (Supplementary Figure 3b).

We then normalized the mutational frequency observed in somatic cancers by comparing them to the number of germ-line variants seen in the human population. To this end, we analyzed variants from over 120,000 individuals (Genome Aggregation Database, gnomAD; www.gnomad.broadinstitute.org; Fig. 1b and Supplementary Figure 2b). This analysis revealed that $R^{6.32}$ shows a relatively low amount of natural variation. Strikingly, by computing the relative variation (i.e. ratio of the frequency of somatic/cancer mutations to that of the germ-line/natural variation; see Methods) for every position, we found that $R^{6.32}$ is selectively the most often mutated position in Class F receptors in cancer genomes compared to the population-level variation (Fig. 1c and Supplementary Figure 2c). As this position is less variable among healthy individuals, but naturally found to be selectively mutated in cancer, these observations suggest that $R^{6.32}$ is likely to be important for physiological receptor activity.

**Contact network between TM6/7 constitutes a molecular switch.** While structural insight into Class F receptors is limited, several crystal structures of SMO provide pertinent information that can be applied to the whole receptor class[29–34]. Detailed investigation into the presence of TM6/7 contacts between residues in the published SMO crystal structures, which represent inactive receptor conformations, indicates that hydrogen bonds and π-cation stacking interactions between $R451^{6.32}$ and the lower end of TM7 ($T534^{7.54}$, $W535^{7.55}$, $W537^{7.57}$) are formed in SMO structures (Fig. 2a, for all residue contact fingerprints between residues in the TM6/7 helices, see Supplementary Figure 4). In addition, the crystal structure of $FZD_4$, the high resolution FZD structure, in the absence of ligand and the extracellular cysteine-rich domain (CRD), also reveals a contact between $K^{6.32}$ and $W^{7.55}$ [35]. In the $FZD_4$ structure, an additional contact between $K^{6.32}$ and $E^{2.41}$—a negatively charged residue only conserved in FZDs—further contributes to the stabilization of this network. Despite the more detailed structural insight into this region in the inactive Class F receptors, it remains obscure what opening of a molecular lock or switch means functionally for signal activation and specification downstream of Class F receptors.

Receptors in a fully active G protein-coupling state undergo an opening of the cytoplasmic cavity of their transmembrane helix bundle to accommodate the α5 helix of the Gα subunit allowing for guanine nucleotide exchange (GEF) activity of the receptor[6]. Along this line, the π-cation and hydrogen bonding interactions

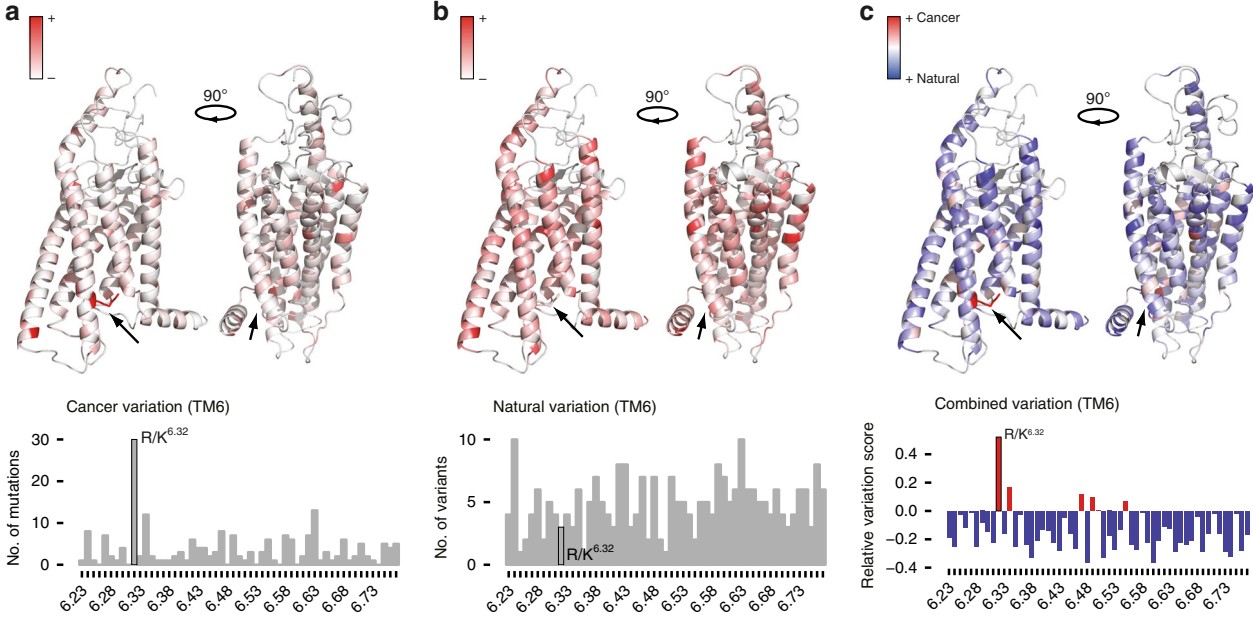

**Fig. 1** A conserved, basic residue in TM6 of Class F receptors is frequently mutated in cancer. **a** Counts of cancer mutations in Class F receptors in human tumors. Color intensity corresponds to mutation frequency. **b** Counts of naturally occurring variants from gnomAD. **c** Relative variation score (see Methods) describing the amount of cancer variation compared to the variability observed in the natural population at each site. Positions where the amount of cancer variation is greater were colored in shades of red, whereas positions with excess of natural variation were colored in shades of blue. Panels **a**, **b**, **c** were generated by projecting sitewise scores onto a $FZD_6$ receptor model. For additional information on mutations in Class F in cancer see Supplementary Figures 2 and 3

of the lock observed between TM6–TM7 in SMO and $FZD_4$ could function as a conserved molecular switch mechanism for ternary complex formation resembling the ionic lock in Class A GPCRs and the recently identified polar network in the Class B GLP-1 receptor[34,36–38]. The analogous mechanism in Class B receptors is also based on an arginine-dependent interaction between the TM6 and TM7/H8, which is broken in the active, G protein-coupled GLP-1 receptor/$G_s$ CryoEM structures (Fig. 2b)[37,39].

Interestingly, one of the tryptophans at the lower end of TM7 ($W^{7.55}$) that is contacted by $R/K^{6.32}$ is conserved in all Class F members (Supplementary Figure 1a) and this residue has been identified as an oncogenic mutant in human SMO (SMOM2; SMOA1 in mouse SMO) mediating PTX-sensitive, $G_i$-dependent glioma-associated oncogene (GLI) transcription factor-mediated transcriptional activation[17,34,40,41]. The mutation of $W^{7.55}$ to L in $FZD_2$, $FZD_6$ and SMO is associated with different forms of cancer (Supplementary Figure 3b). The frequent occurrence of Class F $R^{6.32}$ mutations in human cancers suggests increased activity of mutant receptors similar to the increased constitutive activity of Class A GPCRs upon mutational disruption of their ionic lock[8,42,43] or the residues involved in the structural rearrangement leading to Class A receptor activation[36].

To study the importance of the residue contacts mediated by $R^{6.32}$ in $FZD_6$, a SMO crystal structure (PDB ID: 4JKV) was used as the basis for a $FZD_6$ homology model, where the conserved sites $R416^{6.32}$ and $W493^{7.55}$ are shown juxtaposed in TM6 and TM7, respectively (Fig. 2c). This model reveals hydrogen bonding of the charged $R416^{6.32}$ side chain to oxygen atoms of the TM7 helical backbone and π-cation interactions with the side chain of $W493^{7.55}$ (for models of $FZD_{1-10}$ see Supplementary Figure 5a). Furthermore, computational mutation of position 6.32 reveals that these contacts can neither be formed in the experimental $R416A^{6.32}$ nor the naturally occurring $R416Q^{6.32}$ mutants of $FZD_6$ (Fig. 2c).

We next analyzed the stability of the residue contacts by performing molecular dynamics simulations employing the $FZD_6$ model (Supplementary Figure 5c)[32,44]. In order to more closely characterize the observed changes between wild-type $FZD_6$ and $R416A^{6.32}$, we quantified the distance between TM3-TM6 regions that undergo large conformational changes in Class A GPCRs upon activation[45]. Comparing the distribution of distances between TM3-TM6, the minimum observed distance was smaller in $FZD_6$ than in the $R416A^{6.32}$ mutant. This suggests a higher capability of the wild-type receptor to form a more closed, inactive conformation and the mutant to form a more open, active-like conformation (Fig. 2d). Due to the fact that the MD simulations were carried out in the absence of G protein, the dynamics refer to an intermediate and not fully active state. An additional homology model of $FZD_6$, which is based on the inactive SMO crystal structure fused with the lower part of TM6 modeled according to the active bovine opsin crystal structure in complex with the C5 α-helix of transducin, allowed us to study an active-state conformation including an outward movement of TM6[46]. In this model, the conformational change prevents interactions between $R416^{6.32}$ and TM7—a finding that is consistent with its role as an activation switch (Supplementary Figure 5b). These calculations suggest that mutation of $R^{6.32}$ may facilitate the receptor to sample the active-like conformation more frequently and may confer constitutive basal activation of the receptor in the absence of agonist, but in the presence of the intracellular transducer.

**Mutation of $R^{6.32}$ in $FZD_6$ affects basal receptor activity.** Constitutive activity of GPCRs is traditionally assessed with inverse agonists, where the negative efficacy reduces basal activity in the absence of orthosteric agonist. Due to the inexistence of inverse agonists targeting FZDs, we employed pharmacological inhibitors to create conditions that were free of endogenously secreted WNT proteins in the presence of overexpressed wild type or $FZD_6$ $R416A^{6.32}$ as a means of measuring the ligand-independent, receptor-intrinsic activity. In order to test whether

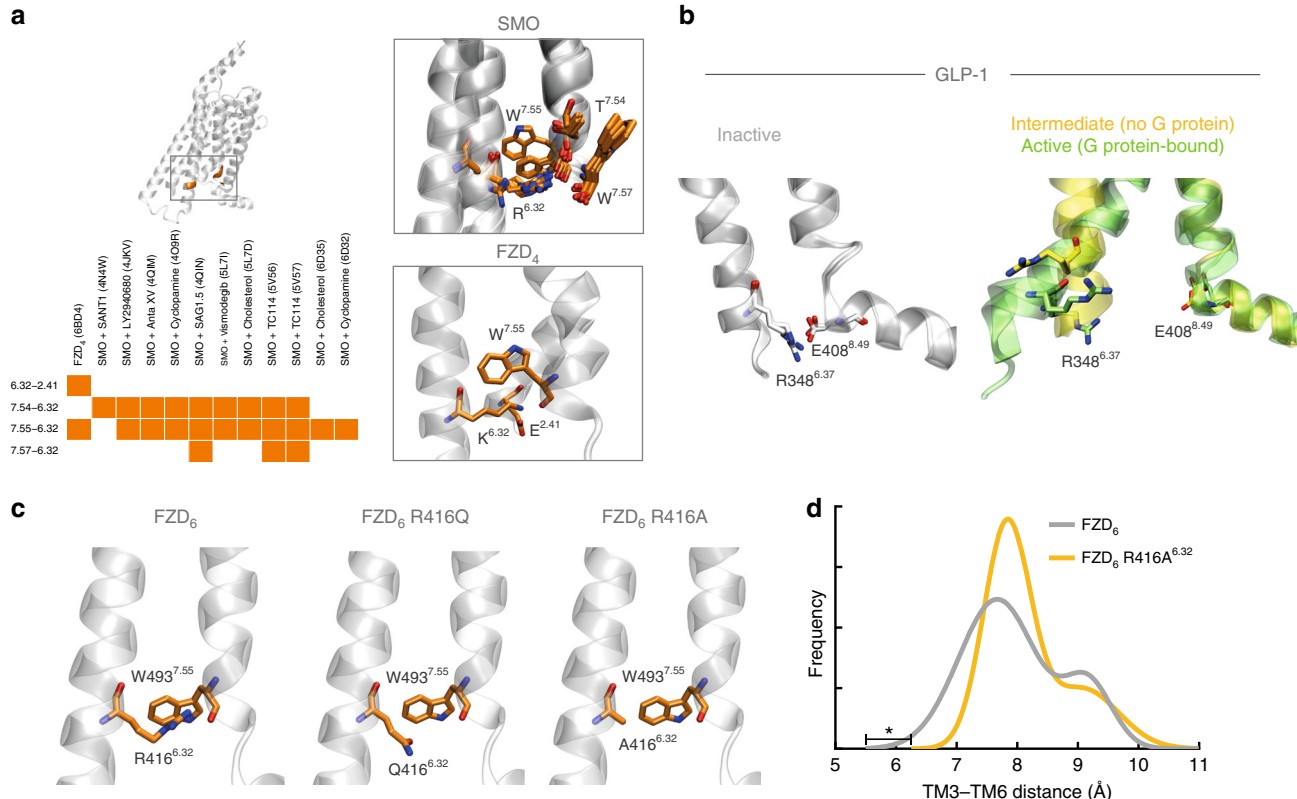

**Fig. 2** Interactions between R/K[6.32] and helix 7 allow for a molecular switch mechanism. **a** General receptor and magnified view centered on residue 6.32 of a structural overlay of all available SMO and FZD[4] crystal structures (PDB IDs SMO: 5L7D, 5L7I, 5V56, 5V57, 4O9R, 4N4W, 4JKV, 4QIM, 4QIN, 6D32, 6D35; FZD[4]: 6BD4). Residue 6.32 and its interacting residues are shown in orange stick representation. The bottom, left inset shows contact fingerprints for all interactions measured using the Protein Contact Atlas between residue 6.32 and residues in TM7 and TM2 (an orange box indicates that the contact is present in that structure, a white box indicates absence of the contact). All structures present inactive structures in the absence of heterotrimeric G protein. **b** Representation of the equivalent receptor region in the previous panel in GLP-1 receptors. Inactive (PDB IDs: 5VEW and 5VEX, gray), intermediate (PDB ID: 5NX2, orange), and active/G protein-bound (PDB IDs: 5VAI and 6B3J, green) structures are shown. The proposed TM6–TM7/ H8 switch residues are shown as sticks. **c** Left panel: computational model of FZD[6] based on the SMO crystal structure (PDB ID: 4JKV). R416[6.32] on TM6 and W493[7.55] on TM7 are highlighted in orange. Middle panel: representation of the naturally occurring cancer mutant FZD[6] R416Q[6.32]. Right panel: representation of the experimental R416A[6.32] mutant. **d** Analysis of the frequency of the of TM3-TM6 (W[3.50]–G[6.34]) distance distributions over MD simulation time for FZD[6] and FZD[6] R416A[6.32]. Threshold values are compared using unpaired $t$ test; $n = 3$ (FZD[6]) $n = 4$ (FZD[6] R416A[6.32]); $P = 0.0281$; $t = 3.060$; df = 5. *$P < 0.05$ (two-tailed $t$-test)

the R416A[6.32] mutation could also confer ligand-independent constitutive activity of exogenously expressed FZD[6], we monitored basal phosphorylation of extracellular-signal regulated kinases 1/2 (ERK1/2)—similar to what we have previously shown[44]. Inhibition of Porcupine—the enzyme that is required for WNT acylation and secretion—blunts endogenous WNT secretion[47]. While HEK293 cells stably expressing FZD[6] exhibited higher basal ERK1/2 phosphorylation compared to control cells, expression of FZD[6] R416A[6.32] was accompanied by a more pronounced ERK1/2 phosphorylation. Incubation with the Porcupine inhibitor C59 reduced both FZD[6]- and FZD[6] R416A[6.32]-induced ERK1/2 phosphorylation. Whereas the wild-type FZD[6] showed a tendency for constitutive activity, FZD[6] R416A[6.32] exhibited a more pronounced constitutive activity in the absence of endogenous WNTs and in the presence of endogenous G proteins (Fig. 3b). These results collectively suggest that mutation of this position confers a higher constitutive activation of the receptor in a ligand-independent manner initiating a cellular response.

**The molecular switch defines functional selectivity of FZDs.** Despite the apparent constitutive activity for the G protein-dependent pathway to ERK1/2[14,44,48], signaling through the

phosphoprotein DVL—a central mediator of WNT/FZD signaling[49]—was negatively affected by disruption of the molecular switch. Both the experimental FZD[6] R416A[6.32] and the naturally occurring cancer mutants of the molecular switch R416Q[6.32] and W493L[7.55] were impaired in the ability to recruit DVL to the membrane and to induce the electrophoretic mobility shift associated with DVL activation (Fig. 4a–c and Supplementary Figure 7)[50,51]. Recruitment of DVL to the plasma membrane was quantified by bystander bioluminescence resonance energy transfer (BRET) employing the Venus-tagged CAAX domain of kras as a membrane marker in combination with an N-terminally Nluc-tagged DVL2. Contrary to the wild-type receptor, all tested mutants of FZD[6] were incapable of recruiting DVL to the membrane as referenced by the negative control, the β[2] adrenergic receptor (Fig. 4a). Furthermore, we took advantage of the recently described phospho-specific antibody detecting the C-terminal, phosphorylated S648 of FZD[6], which is indicative of functional casein kinase 1 (CK1) targeting and DVL recruitment[52]. While FZD[6] is significantly phosphorylated in the presence of coexpressed CK1ε and DVL2, disruption of the molecular switch in all three mutants impaired S648 phosphorylation, leaving FZD[6] W493L[7.55] with residual S648 phosphorylation (Fig. 4d, e).

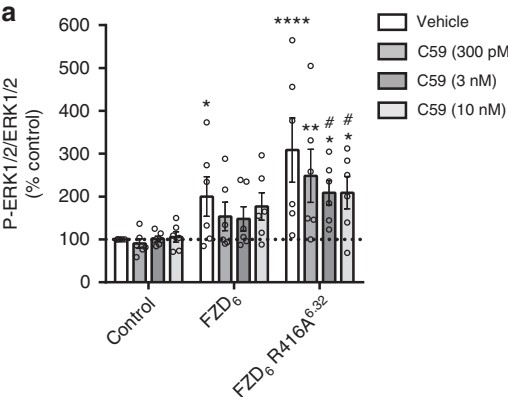

**Fig. 3** Mutation of $R^{6.32}$ in $FZD_6$ affects basal receptor activity. $FZD_6$- and $FZD_6$ $R416A^{6.32}$-induced ERK1/2 phosphorylation in the absence and presence of the Porcupine inhibitor C59 (300 pM, 3 nM, 10 nM) in HEK293 cells stably expressing the receptors. P-ERK1/2 and total ERK1/2 levels were quantified by multiplex AlphaScreen. Data are presented as mean ± standard error of the mean (s.e.m.). $n = 6$; $P = 0.0002$, $F$ (23, 119) = 2.718. $*P < 0.05$, $**P < 0.01$, $****P < 0.0001$ represent comparisons of receptor-mediated P-ERK1/2/ERK1/2 levels with vehicle-treated control cells. $\#P < 0.05$ represents the comparison of P-ERK1/2/ERK1/2 levels in C59-treated cells with vehicle-treated cells (one-way ANOVA)

Because $FZD_6$ is more restrictive in its pathway selectivity and is not known to mediate WNT/β-catenin signaling[53], we extended our studies to $FZD_5$, which is known to mediate both G protein- and WNT/β-catenin-dependent signaling[25,54]. Similar to $FZD_6$, mutation of the molecular switch in $FZD_5$ abolished DVL recruitment to the membrane to levels comparable to the $β_2$ adrenergic receptor (Fig. 4f). In agreement with a loss in FZD-DVL interaction, $FZD_5$ $R449A^{6.32}$ was not able to mediate WNT-3A-induced T cell factor (TCF)/lymphocyte enhancer factor (LEF)-dependent transcriptional activity as monitored by the TOPFlash assay in cells devoid of endogenous $FZD_{1-10}$ expression (Fig. 4g)[55]. Given the lack of endogenous FZDs, WNT-3A stimulation did not evoke a response in control-transfected cells. While $FZD_5$ expression dramatically enhanced the TCF/LEF transcriptional activity in response to WNT-3A compared to the empty vector control, $FZD_5$ $R449A^{6.32}$ did not. In order to exclude the possibility that the absence of a response in cells transfected with the mutant receptor might be due to poor membrane expression of SNAP-$FZD_5$ $R449A^{6.32}$, we optimized transfection to achieve similar receptor surface levels validated by flow cytometry (Supplementary Figure 6a) using a cell impermeable, fluorescent SNAP substrate in parallel to the TOPFlash experiments. Transfection conditions that yielded similar surface expression of the receptor in HEK293 cells were compared for the ability to mediate WNT-3A-induced TCF/LEF transcriptional activity in the cells lacking $FZD_{1-10}$, clearly underlining the inability of the SNAP-$FZD_5$-$R449A^{6.32}$ to mediate WNT/β-catenin signaling.

Collectively, these findings with $FZD_5$ and $FZD_6$ merge well with the current understanding of the existence of different ternary complexes defined by the nature of the intracellular transducer[56] and the concept of functional selectivity or signaling bias[57]. The $FZD_6$ $R416A^{6.32}$ mutation preferentially accommodates G protein binding over DVL interaction as evidenced by the ability of the mutant receptor to induce P-ERK1/2 and its inability to induce PS-DVL or to recruit DVL to the membrane (Fig. 4a–d and Supplementary Figure 7). Conversely, our data suggest that wild-type $FZD_6$ could be biased towards interaction with DVL over heterotrimeric G protein—a process that could be

affected by local differences in transducer concentrations. In this context, the inability of $FZD_5$ $R449A^{6.32}$ to recruit DVL and to mediate WNT/β-catenin signaling supports this model. Previous studies on $FZD_4$ identified a mutation at the lower end of TM2, at the evolutionary conserved $Y250^{2.39}$, which negatively affects DVL interaction while maintaining its ability to functionally interact with heterotrimeric $G_{12/13}$ proteins[58]. In contrast, the $FZD_6$ R511C nail dysplasia mutant maintained interaction with DVL, but lost its ability to associate with $G_i$ or $G_q$[14]. Together with our current findings, these data collectively support the existence of distinct conformational states that selectively feed into either DVL or heterotrimeric G protein signaling (Fig. 4h).

**mG sensors detect a fully active Class F receptor state.** In order to better understand the mechanism of action of the $R/K^{6.32}$ mutations present in Class F receptors and given the absence of a high resolution ternary complex structure, we made use of recently developed conformational sensors of GPCR activation—so-called mG proteins. These mG proteins have served to detect the active state conformation of GPCRs in living cells and to stabilize active, purified receptors for crystallization and CryoEM studies[20–24]. These engineered G proteins were fused to Venus to serve as BRET acceptors in combination with C-terminal luciferase-tagged Class F receptors as energy donors (Fig. 5a). Based on emerging evidence that Class F receptors function as bona fide GPCRs[1,12,13,15–18,59,60] and similar to what was shown before for the use of Venus-tagged mG proteins in combination with Class A GPCRs[24], we postulated that agonist stimulation of Class F receptors would lead to the recruitment of the mG protein to the receptor.

The ten FZDs are subdivided into four evolutionarily-related clusters consisting of $FZD_{1, 2, 7}$, $FZD_{3, 6}$, $FZD_{5, 8}$, and $FZD_{4, 9, 10}$ (Fig. 5b). With the aim of investigating the generality of the presented mechanism, we assessed mG protein interaction with one representative of each FZD homology cluster and SMO. Based on what is known about FZD-G protein selectivity, we focused on $FZD_4$-$G_{13}$[61], $FZD_5$-$G_q$[54], $FZD_6$-$G_i$[14,59], $FZD_7$-$G_s$[62], and SMO-$G_i$[17,18,63]. BRET assays were performed in transiently transfected HEK293 cells using recombinant, purified WNT-5A (FZDs) and SAG (SMO) as agonists. Concentration-response curves were produced comparing the potency of agonist at the wild-type receptor with the $R/K^{6.32}$ to alanine mutants. A dramatic left shift in the agonist potency was detectable for all tested $R/K^{6.32}$ Class F mutants compared to the respective wild-type receptors at similar surface expression levels (Fig. 5c–g; Supplementary Figure 6b, c). In addition to the experimental $R/K^{6.32}$ to alanine mutants, we have also performed mG BRET experiments using the naturally occurring cancer mutants $FZD_6$ $R416Q^{6.32}$ and SMO $R455H^{6.32}$, as well as $FZD_6$ $W493L^{7.55}$ and SMO $W539L^{7.55}$ (Supplementary Figures 6d, 8a, b). In short, these experiments confirmed that: (1) the validated mG proteins[24] act as conformational sensors, detecting and binding to the active conformation of the respective Class F receptors, (2) the mutation of $R/K^{6.32}$ or $W^{7.55}$ increases the potency of agonists by being able to bind better to the cognate G proteins and (3) the naturally occurring cancer mutants in the molecular switch mechanistically phenocopy the experimental alanine mutants. In order to further complement our conclusions, we ran MD simulations of SMO and its naturally occurring cancer mutants $R^{6.32}$ to $H^{6.32}$ and $W^{7.55}$ to $L^{7.55}$ based on the crystal structure of human SMO in the absence of the extracellular CRD and without a crystallization scaffold in IL3 (PDB 4JKV; Supplementary Figure 8c–f). For the time of the MD simulation (150 ns, 3 replicates), the positioning of the residues was more stable and TM6/7 interactions in the molecular switch region were more

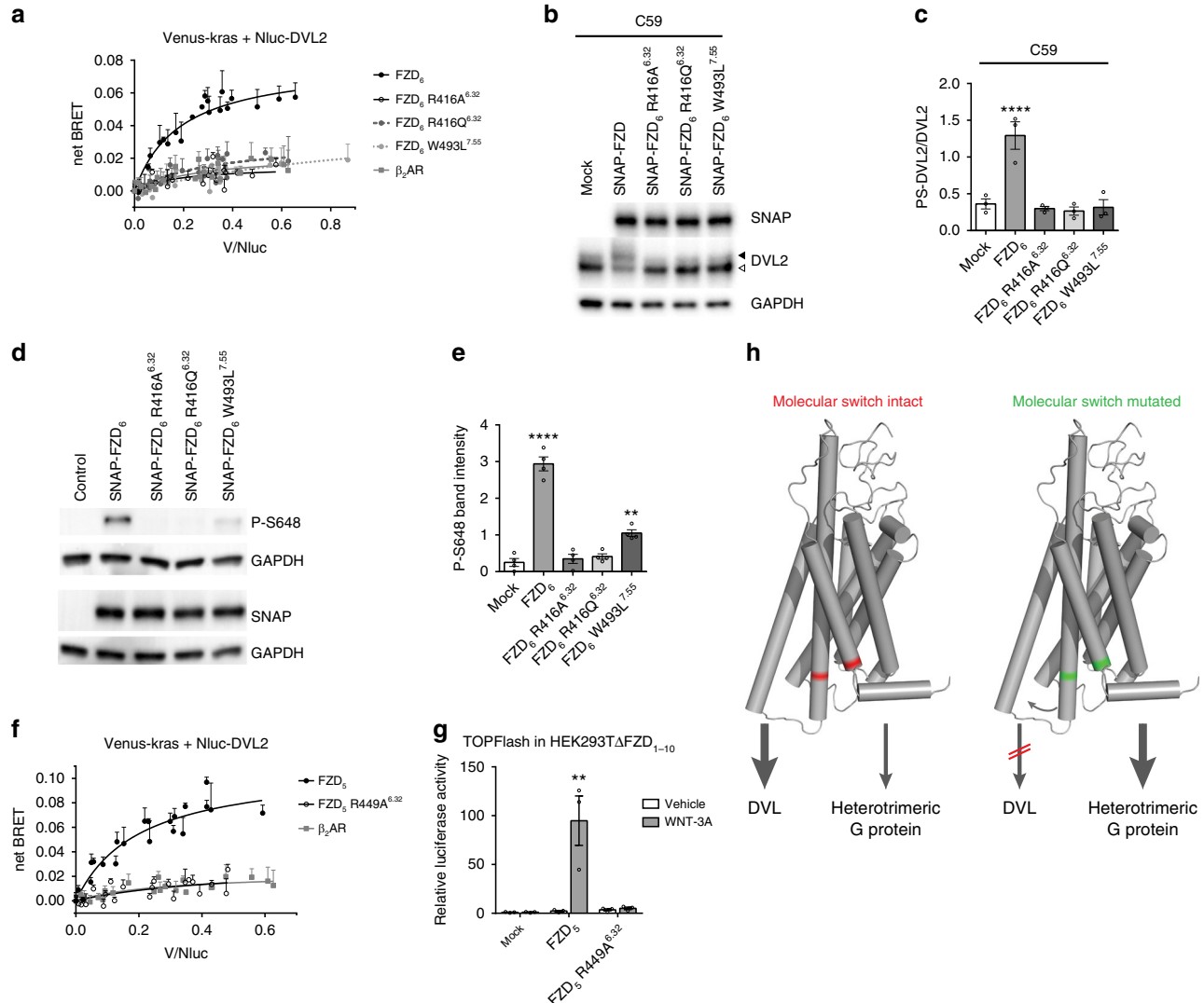

**Fig. 4** Mutation of the molecular switch confers functional selectivity. **a** To quantify DVL2 membrane recruitment, bystander BRET between Venus-kras and Nluc-DVL2 was assessed over a range of acceptor/donor ratios in the presence of $FZD_6$, $FZD_6$ $R416A^{6.32}$, $FZD_6$ $R416Q^{6.32}$, $FZD_6$ $W493L^{7.55}$ or the $\beta_2$-adrenergic receptor as negative control. net BRET values are presented as mean ± standard deviation (s.d.) of $n = 3$ independent experiments. **b**, **c** HEK293 cells transfected with empty vector (control), $FZD_6$, $FZD_6$ $R416A^{6.32}$, $FZD_6$ $R416Q^{6.32}$, or $FZD_6$ $W493L^{7.55}$ were analyzed by immunoblotting using anti-SNAP, -DVL2, and -GAPDH (loading control) antibodies. Bar graphs for the ratio of PS-DVL2 (upper band) to DVL2 (lower band) summarize densitometry data. Experiments were performed in the presence of 5 nM C59 (overnight). Data are presented as mean ± s.e.m. of $n = 4$ independent experiments; $P = 0.0002$, $F (4, 10) = 17.71$. ***$P < 0.001$ (one-way ANOVA). See also Supplementary Figure 7. **d**, **e** HEK293 cells cotransfected with empty vector, $FZD_6$, $FZD_6$ $R416A^{6.32}$, $FZD_6$ $R416Q^{6.32}$, or $FZD_6$ $W493L^{7.55}$ and DVL2/CK1ε were analyzed by immunoblotting using anti-phospho-S648 $FZD_6$, anti-SNAP, and anti-GAPDH antibodies. The P-S648 signal was quantified by densitometry and summarized in a bar graph. Data are presented as mean ± s.e.m. of $n = 3$ independent experiments; $F (4, 15) = 83.78$., ****$P < 0.0001$, **$P < 0.01$ (one-way ANOVA). **f** In a similar setup to **a** bystander BRET was measured between Venus-kras and Nluc-DVL2 in the presence of $FZD_5$, $FZD_5$ $R449A^{6.32}$ or the $\beta_2$-adrenergic receptor (data points for $\beta_2$-adrenergic receptor are identical to **a**). net BRET values are presented as mean ± s.d. of $n = 3$ independent experiments. **g** HEK293T$\Delta$FZD$_{1-10}$ cells were transfected with Renilla and Firefly luciferase together with empty vector (control), $FZD_5$, or $FZD_5$ $R449A^{6.32}$ and stimulated with 300 ng ml$^{-1}$ recombinant WNT-3A overnight. The luciferase signal was normalized to the average of unstimulated control values. Data are represented as mean ± s.e.m. of $n = 3$ independent experiments. $P = 0.0065$, $F (2, 6) = 13.12$. **$P < 0.01$ (one-way ANOVA). **h** Schematic presentation of the concept of conformation-driven signaling bias of wild-type FZD and molecular switch mutant FZD. FZD models were produced in PyMOL (The PyMOL Molecular Graphics System, Version 2.0 Schrödinger, LLC)

long-lived in the wild type than in the mutant receptors. These in silico observations support the concept of a molecular switch in receptor activation allowing association of an intracellular transducer—the heterotrimeric G protein.

Due to the general lack of well-characterized small molecule drugs targeting FZDs, we employed additional compounds acting at SMO to characterize the mode of action of the $R/K^{6.32}$ molecular switch. Similar to what was previously observed with SAG in a luciferase-based reporter assay[64], the agonist presented

a bell-shaped concentration-response curve in the mG protein recruitment assay. It was suggested that SAG acts on an off-target site at higher concentrations and this is supported by the finding that cyclopamine-KAAD, an orthosteric inverse agonist, solely affects the SAG concentration-response curve on the ascending part of the bell-shaped curve[40,64]. In the SMO-mGsi BRET assay, the SAG concentration-response curves in wild type and $R455A^{6.32}$ SMO appear similarly biphasic. Incubation with 100 nM cyclopamine-KAAD reversed the $R455A^{6.32}$ mutation

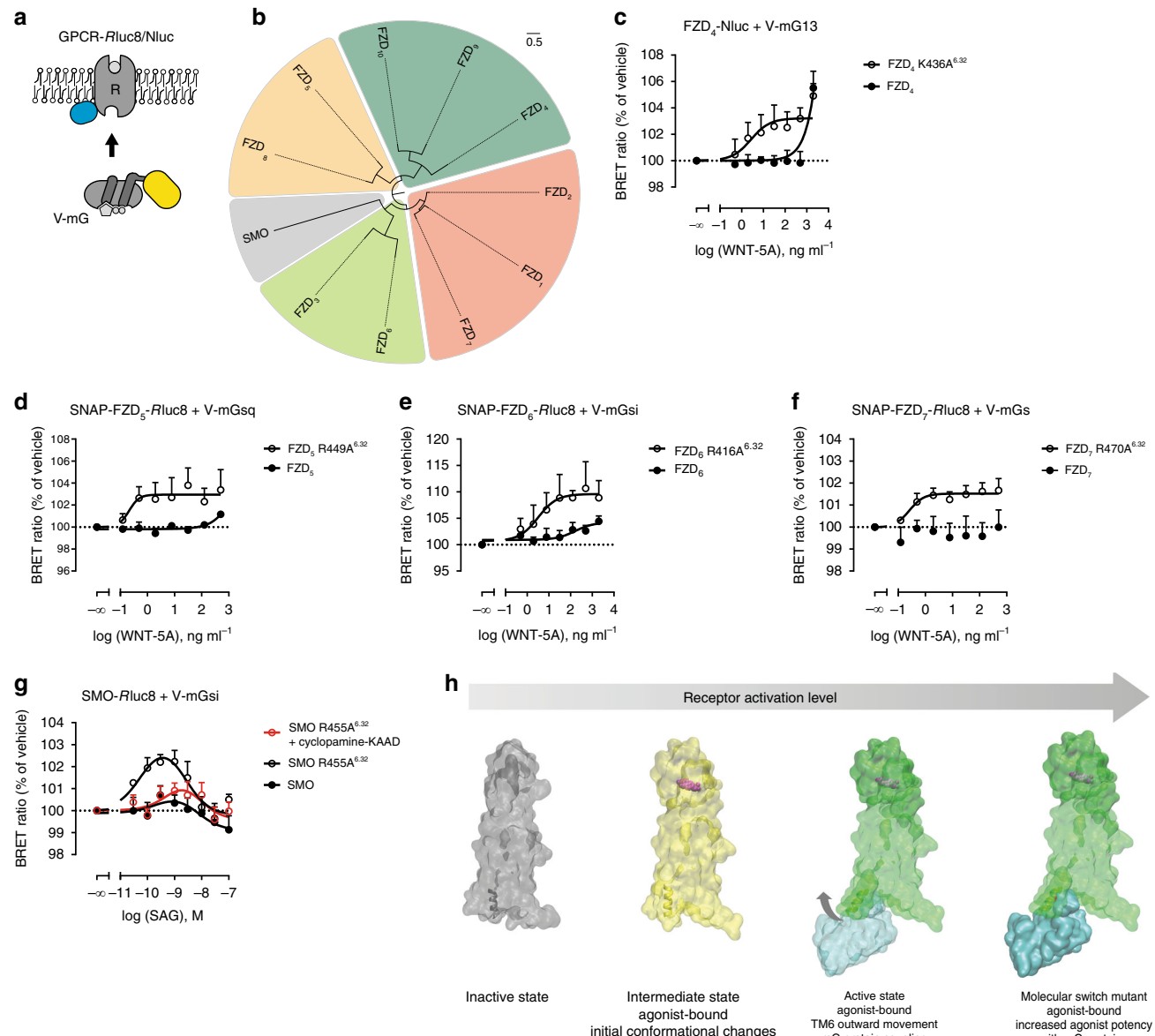

**Fig. 5** Switch mutations at position 6.32 of Class F receptors increase agonist potency. **a** Illustration depicts the experimental setup wherein luciferase-tagged Class F receptors (R) are expressed at the plasma membrane and validated Venus-tagged mG proteins are localized to the cytosol[24]. Relying on excitation from proximity to the luciferase (donor), the Venus-tagged mG protein (acceptor) fluoresces only when the receptor is in its active conformation. **b** Maximum-likelihood phylogenetic tree of human FZD and SMO paralogs, with the four major FZD clusters and SMO color-coded. Branch lengths are given in amino-acid substitutions per site. **c–g** BRET experiments in HEK293 cells transiently expressing representatives of Class F with mG proteins: **c** $FZD_4$/mG13 ($n = 6$), **d** $FZD_5$/mGsq ($n = 5$), **e** $FZD_6$/mGsi (wild type $n = 7$; R416A[6.32] $n = 6$), **f** $FZD_7$/mGs ($n = 8$), and **g** SMO/mGsi ($n = 6$). Wild-type receptor (filled circle) and molecular switch mutants (open circle) were compared in parallel and receptor surface expression was measured by bystander BRET and flow cytometry (Supplementary Figure 6). Cells were stimulated with the indicated concentrations of recombinant WNT-5A or SAG and the normalized BRET ratio of Venus to Rluc8/Nluc was measured. In **g**, effects of SAG alone were compared to increasing concentrations of SAG in the presence of the inverse agonist cyclopamine-KAAD (100 nM; red open circle). Data are represented as mean ± s.e.m. **h** Summary scheme illustrating the activation states of Class F receptors in the absence and presence of receptor-activating ligands, mG protein and the R/K[6.32] mutation. Only the combination of agonist and mG protein can stabilize a fully active state. The receptor models in the active state are a fusion of the full-length SMO structure (PDB ID 6D35) and the lower end of TM6 of the adenosine $A_{2A}$ receptor in complex with a mG protein (PDB ID 5G53[20])

phenotype in SMO, shifting the curve rightward comparable to the wild-type SMO without affecting the descending segment of the curve (Fig. 5g).

Given the distinct differences between wild type and the R/K[6.32] or W[7.55] mutants, it could be possible that mutation of the molecular switch region conveys the ability to couple to heterotrimeric G proteins promiscuously. In order to exclude this possibility, we examined the G protein-coupling profile of

wild-type SMO in a nucleotide depletion assay allowing to directly assess the formation of a ternary complex by BRET in the absence of nucleotides. Constitutive activity or ligand-independent G protein-coupling cannot be measured with mG proteins and so we made use of full heterotrimeric G proteins, which are nucleotide sensitive in order to define the constitutive activity of wild-type SMO towards heterotrimeric G proteins. To this end, we created conditions where the G protein would

have a higher affinity for the receptor by removing GDP and GTP through apyrase treatment in permeabilized cells. We then promoted the dissociation of the heterotrimeric G protein through the addition of the inverse agonist cyclopamine. The difference, reflected by the decrease in BRET between luciferase-tagged wild-type SMO, Venus-tagged Gβγ, and untagged Gα, in the presence or absence of cyclopamine revealed that SMO couples to $G_i$ and $G_{12}$, but not to $G_q$ or $G_s$ (Supplementary Figure 9a)—in agreement with previously published results[17,18,65]. Using mG proteins to control for the G protein specificity of SMO R455A[6.32], we confirmed that mutation of the molecular switch does not render the receptor promiscuous (Supplementary Figure 9b).

## Discussion

Our data identify a conserved network of interactions in TM6/TM7, which serves as a molecular switch required for the full activation of G protein-bound Class F receptors. These findings contribute to a better understanding of Class F receptor activation mechanisms connecting structural indications[34] with functional signaling output in a family-wide approach using large scale genomic data analysis, bioinformatics, and functional readouts including conformational mG protein sensors. Furthermore, our data suggest the existence of conformational bias in signal initiation and specification, partitioning signaling through heterotrimeric G proteins and the phosphoprotein DVL to distinct receptor complexes that depend on biased receptor conformational states. This concept is well-established in the field of GPCR pharmacology,, where exciting opportunities for the development of biased ligands promise improved selectivity and reduced unwanted side effects[66]. More work needs to be done to structurally define the distinct receptor conformations and structural features in Class F receptors that define coupling selectivity to different transducer proteins, such as heterotrimeric G proteins and DVL. However, these findings merge well with previous data showing that overexpression of DVL negatively impacts FZD-G protein interaction and signaling[14,16]. Based on different signaling profiles of purified WNTs in FZD-expressing mouse microglia-like N13 cells or FZD-free 32D cells stably expressing individual FZD isoforms, we had proposed that WNTs could act as biased ligands of FZDs distinguishing G protein over DVL signaling, even though this interpretation still needs to be pharmacologically and quantitatively validated[67,68].

Mutations in W[7.55] in SMO, a residue that we define here as part of the Class F molecular switch, were previously identified as oncogenic drivers[40,41]. Despite the fact that the R[6.32] is the most frequently mutated residue in FZDs in human cancers, it remains obscure if and how mutations in the molecular switch (Supplementary Figure 3) render FZDs oncogenic. While the mutated molecular switch in FZDs apparently does not provide input to the DVL-dependent WNT/β-catenin pathway, enhanced FZD-induced activation of heterotrimeric G proteins could provide tumor-promoting signals[8]. Since the present study employs cancer and population genomics solely to identify residues of mechanistic importance for receptor activation, further studies are required to define the contribution to and the underlying mechanisms of molecular switch mutations in Class F receptors found in human cancer.

In summary, our findings open the door for the development of high-throughput-compatible screening assays directly monitoring Class F receptor activation on a structural level instead of using signal amplified transcriptional reporter assays that are prone to deliver off-target hits[69]. Moreover, our data are directly applicable to mechanism-based drug discovery and the potential

development of biased compounds targeting abnormal Class F receptor-mediated G protein signaling in cancer. Drugs such as cyclopamine-KAAD that target oncogenic mutants of SMO display an effect on the R[6.32] molecular switch providing the proof-of-principle that FZDs may also be targeted in a similar way to combat diseases associated with upregulated WNT/FZD signaling[40].

## Methods

**Computational modeling and molecular dynamics simulation**. The homology models of inactive $FZD_{1-10}$ were generated using a structure of SMO as a template (PDB ID: 4JKV)[32]. The sequences of $FZD_1$ (UniProt ID: Q9UP38), $FZD_2$ (UniProt ID: Q14332), $FZD_3$ (UniProt ID: Q9NPG1), $FZD_4$ (UniProt ID: Q9ULV1), $FZD_5$ (UniProt ID: Q13467), $FZD_6$ (UniProt ID: O60353), $FZD_7$ (UniProt ID: O75084), $FZD_8$ (UniProt ID: Q9H461), $FZD_9$ (UniProt ID: O00144), and $FZD_{10}$ (UniProt ID: Q9ULW2) were aligned to that of SMO (UniProt ID: Q99835) with ClustalX2[70]. The N- and C-termini were excluded due to a lack of suitable template and the alignment was manually edited to ensure the proper alignment of transmembrane domains and conserved motifs present in Class F GPCRs. In order to generate an active-like $FZD_6$ model, we used the crystal structure of rhodopsin, which is also a $G_i$-coupled receptor, in its G protein-bound conformation as a template (PDB ID: 3DQB)[46]. Residues 408–427 ($E^{6.24}$–$P^{6.43}$) from TM6 of $FZD_6$ were modeled using corresponding residues from TM6 ($A^{6.24}$–$A^{6.43}$) of rhodopsin. Fifteen homology models of each FZD receptor were generated with MODELLER 9.19[71] and the representative ones were selected based on DOPE score and visual inspection. $R416^{6.32}$ of $FZD_6$ was mutated to $A416^{6.32}$ in UCSF Chimera 1.11.2 software[72].

Information about Class F receptor mutations in human tumor samples was extracted from the cBioPortal for Cancer Genomics[73]. In order to systematically characterize contacts of residue 6.32 in all SMO crystal structures (PDB IDs: 5L7D, 5L7I, 5V56, 5V57, 4O9R, 4N4W, 4JKV, 4QIM, and 4QIN), we retrieved interhelical contacts using the Protein Contact Atlas with default conditions[74]. In order to filter contacts between consecutive residues, we disregarded all contacts that were 4 or less amino acids apart in the receptor sequence. The GPCRdb was then used to annotate the detected interactions according to Ballesteros—Weinstein numbering. For a complete list of all calculated interaction fingerprints please refer to Supplementary Figure 4. In order to compare and visualize all SMO structures, we superposed all the aforementioned PDB crystal structures in VMD 1.9.4 using STAMP implemented in the MultiSeq extension[75,76]. The same approach was followed to superpose GLP-1 receptor structures in their inactive (PDB IDs: 5VEW and 5VEX), their intermediate (PDB ID: 5NX2), and their activated (PDB IDs: 5VAI and 6B3J) forms.

MD simulations were performed using the NAMD 2.12 simulation package[77]. The inactive $FZD_6$ and $FZD_6$ $R416A^{6.32}$ models were placed in hydrated 1-palmitoyl-2-oleoyl-sn-glycero-3-phosphocholine (POPC) lipid bilayer. The system was solvated in water and its charge neutralized with NaCl. The CHARMM36 force field[78] was used for proteins and lipids, TIP3P model was used for water molecules and NBFIX parameters were used for $Na^+$ and $Cl^-$ ions. The system was minimized in 100000 steps. Subsequently, the system was heated up to 310 K and the POPC lipid bilayer equilibrated for 1 ns with other system components fixed. In order to gradually equilibrate the system, four 250 ps equilibration simulations were run. Harmonic constraints were applied on protein, protein backbone and Cα atoms, respectively. The protein was released in the last equilibration simulation. Three ($FZD_6$) and four ($FZD_6$ $R416A^{6.32}$) independent, unrestrained 235–285 ns NPT ensemble production simulations were run for each receptor. A time step of 2 fs was used. The temperature at 310 K was kept with Langevin dynamics and pressure at 1 bar was held with Nose-Hoover Langevin piston. Particle-mesh Ewald for electrostatic interactions and a 9 Å cut-off for van der Waals interactions were used. Water bond lengths and angles were constrained using SETTLE algorithm and for other molecules, bonds between hydrogens and other atoms were constrained using SHAKE algorithm.

Additionally, MD simulations were performed on the inactive human SMO and cancer-associated $R451H^{6.32}$ and $W535L^{7.55}$ mutant structures using GROMACS 2016.4[79]. The crystal structure of an inactive SMO with an intact IL3 (PDB ID: 4JKV) was downloaded from www.rcsb.org and missing residues (351–354, 494–506) modeled in Modeller using the full-length SMO structure (PDB ID: 5L7D) as a template. Structures of the mutants were generated and protonation states assigned at pH = 7.4 in Chimera. CHARMM-GUI server[80] was used to embed the proteins in the POPC lipid bilayer, add water molecules and 0.15 M NaCl. The system was minimized in 1500 steps and was subsequently subjected to equilibration with gradually-decreasing position restraints on protein and lipid components. In the last 50 ns of the equilibration run, the harmonic force constants of 50 kJ mol$^{-1}$ nm$^{-2}$ were applied on the protein atoms. Lastly, three independent 150 ns isobaric and isothermic (NPT) ensemble production simulations for each receptor were initiated from random velocities. In these simulations, the CHARMM36m force field[81] was used with a 2 fs-time step. The temperature at 310 K was maintained with Nose-Hoover thermostat and the pressure at 1 bar was maintained with Parinello Rahman bariostat. Particle-mesh Ewald for electrostatic interactions and a 9 Å cut-off for

van der Waals interactions were used. All the bonds between hydrogen and other atoms were constrained using the LINCS algorithm. The data files were saved every 100 ps. The MD simulation data (~3 μs combined) were analyzed using VMD and PyMol (The PyMOL Molecular Graphics System, Version 2.0 Schrödinger, LLC).

**Cell culture and transfections.** HEK293 cells (ATCC) were cultured in DMEM supplemented with 10% FBS, 1% penicillin/streptomycin, and 1% L-glutamine (all from Invitrogen Technologies) in a humidified $CO_2$ incubator at 37 °C. All cell culture plastics were from Sarstedt, unless otherwise specified. Pharmacological inhibition of SMO was accomplished with cyclopamine-KAAD (Abcam). C59 (2-[4-(2-Methylpyridin-4-yl)phenyl]-N-[4-(pyridin-3-yl)phenyl]acetamide; Abcam) was used to inhibit Porcupine to abrogate endogenous secretion of WNTs. For stimulation, recombinant WNT-5A (645-WN; R&D Systems/Biotechne) and SAG (N-Methyl-N′-(3-pyridinylbenzyl)-N′-(3-chlorobenzo[b]thiophene-2-carbonyl)-1,4-diaminocyclohexane; Abcam) were used.

In order to generate cell lines stably expressing SNAP-FZD$_6$ and SNAP-FZD$_6$ R416A$^{6.32}$, HEK293 cells were transfected with SNAP-FZD$_6$ or SNAP-FZD$_6$ R416A$^{6.32}$ constructs using Lipofectamine 2000 (Thermo Fisher Scientific), according to the manufacturer's instructions. About 24 h post transfection cells were passaged at 1:10 and 48 h post transfection medium was supplemented with 300 μg ml$^{-1}$ zeocin (Thermo Fisher Scientific). The medium was replaced every two days to select the cells transfected with the plasmids. The cells were maintained in the presence of the antibiotic for a period of 4 weeks until the stable culture was established. Monoclonal cell populations were isolated by limiting dilution. HEK293 control cells underwent the same selection procedure. The stability of protein expression and homogeneity of cell population were verified by immunoblotting and flow cytometry. The stable cell lines were maintained in complete DMEM medium in the presence of 150 μg ml$^{-1}$ zeocin. Absence of mycoplasma contamination was routinely confirmed by PCR using 5′-ggc gaa tgg gtg agt aac acg-3′ and 5′-cgg ata acg ctt gcg act atg-3′ primers detecting 16S ribosomal RNA of mycoplasma in the media after 2–3 days of cell exposure.

**Cloning of receptor constructs and mutagenesis.** FLAG-SNAP-β$_2$AR was from Davide Calebiro (University of Birmingham, UK). hFZD$_4$-Nluc was subcloned from hFZD$_4$-EGFP (Robert J. Lefkowitz, Duke University, USA) into pNluc-N1 with BamHI and NheI. The mouse SMO coding sequence was amplified from pEGFP-mSmo (Addgene plasmid #25395) with primers incorporating a 5′ HindIII site and a 3′ EcoRI site, and subcloned into pRluc8-N1. Mouse SMO forward primer: 5′-atc gct agc gct aaa gct tgc cac cat ggc cgc tgg ccg ccc cgt gcg tgg g-3′. Mouse SMO reverse primer: 5′-tac cgt cga ctg cag aat tcc gaa gtc cga gtc tgc atc caa gat ctc-3′. SNAP-FZD$_5$ and SNAP-FZD$_6$ were from Madelon M. Maurice (Utrecht University Medical Center, The Netherlands) and SNAP-FZD$_7$ was from Ali Jazayeri (Heptares Therapeutics, London, UK). All SNAP-tagged FZDs were cloned into Rluc8-N1 using the following primers and inserted with HindIII and AgeI restriction sites. SNAP-FZD$_5$ forward primer: 5′-gac aag ctt gcc acc atg gtc ccg tgc acg ctg ctc ctg-3′. SNAP-FZD$_5$ reverse primer: 5′-cgt acc ggt gct acg tgc gac agg gac act tgc ttg tgg tat gc-3′. SNAP-FZD$_6$ forward primer: 5′-gac aag ctt gcc acc atg gtc ccg tgc acg-3′. SNAP-FZD$_6$ reverse primer: 5′- cgt acc ggt gca gta tct gaa tga caa cca cct ccc tgc tct tt-3′. SNAP-FZD$_7$ forward primer: 5′-gac aag ctt gcc acc atg gcc tta cca gtg acc gcc ttg ctc ct-3′. SNAP-FZD$_7$ reverse primer: 5′-cgt acc ggt gca tgg tga tgg tga tgg tga tgg tga tgg tga tga tct-3′. Nluc-mDVL2 was subcloned from mDVL2 (Mariann Bienz, MRC, UK) into pNluc-C1 with HindIII and BamHI.

R/K$^{6.32}$ and W$^{7.55}$ mutants were made using QuikChange (Agilent) or Geneart (Invitrogen A13282) with the following primers: FZD$_4$ K436A$^{6.32}$-Nluc forward primer: 5′-agt tag aaa gac tga tgg tcg cga ttg ggg tgt tct cag tac-3′. FZD$_4$ K436A$^{6.32}$-Nluc reverse primer: 5′-gta ctg aga aca ccc caa tcg cga cca tca gtc ttt cta act-3′. SNAP-FZD$_5$ R449A$^{6.32}$ and SNAP-FZD$_5$ R449A$^{6.32}$-Rluc8 forward primer: 5′-gag aag ctc atg atc gcc atc ggc atc ttc ac-3′. SNAP-FZD$_5$ R449A$^{6.32}$-Rluc8 reverse primer: 5′-gtg aag atg ccg atg gcg atc atg agc ttc tc-3′. SNAP-FZD$_6$ R416A$^{6.32}$ and SNAP-FZD$_6$ R416A$^{6.32}$-Rluc8 forward primer: 5′- acc aag aaa aac taa aga aat tta tga ttg caa ttg gag tct tca gcg gctt-3′. SNAP-FZD$_6$ R416A$^{6.32}$ and SNAP-FZD$_6$ R416A$^{6.32}$-Rluc8 reverse primer: 5′-aag ccg ctg aag act cca att gca atc ata att tct ttt agt ttt tct tgg t-3′. SNAP-FZD$_6$ R416Q$^{6.32}$ and SNAP-FZD$_6$ R416Q$^{6.32}$-Rluc8 forward primer: 5′-aga aat tta tga ttc aac tgg agt ctt ca g-3′. SNAP-FZD$_6$ R416Q$^{6.32}$ and SNAP-FZD$_6$ R416Q$^{6.32}$-Rluc8 reverse primer: 5′-ctg aag act cca att tga atc ata att tc t-3′. SNAP-FZD$_6$ W493L$^{7.55}$ and SNAP-FZD$_6$ W493L$^{7.55}$-Rluc8 forward primer: 5′-atc tct gct gtc ttc ctg gtt gga agc aaa aa-3′. SNAP-FZD$_6$ W493L$^{7.55}$ and SNAP-FZD$_6$ W493L$^{7.55}$-Rluc8 reverse primer: 5′-ttt ttg ctt cca acc agg aag aca gca gag at-3′. SNAP-FZD$_7$ R470A$^{6.32}$-Rluc8 forward primer: 5′-gag aag ctc atg atg gtg gcc atc ggc ttc ag-3′. SNAP-FZD$_7$ R470A$^{6.32}$-Rluc8 reverse primer: 5′-ctg aag acg ccg atg gcc acc atg agc ttc tc-3′. SMO R455A$^{6.32}$-Rluc8 forward primer: 5′-caa cga gac cat gct ggc cct ggg cat ttt tgg c-3′. SMO R455A$^{6.32}$-Rluc8 reverse primer: 5′-gcc aaa aat gcc cag ggc cag cat ggt ctc gtt g-3′. SMO R455H$^{6.32}$ forward primer: 5′-acg aga cca tgc tgc acc tgg gca ttt ttg g-3′. SMO R455H$^{6.32}$ reverse primer: 5′-cca aaa atg ccc agg tgc agc atg gtc tcg t-3′. SMO W539L$^{7.55}$ forward primer: 5′- att gcc atg agc acc ctg gtc tgg acc aag gc-3′. SMO

W539L$^{7.55}$ reverse primer: 5′- gcc ttg gtc cag acc agg gtg ctc atg gca at-3′. All constructs were confirmed by sequencing.

**Bioluminescence resonance energy transfer (BRET) assays.** HEK293 cells were transiently transfected in suspension using Lipofectamine 2000 and seeded onto poly-D-lysine (PDL)-coated white or black 96-well cell culture plates with solid f-bottom (Greiner Bio-One). About 48 h post transfection, cells were washed once with BE buffer (150 mM NaCl, 2.5 mM KCl, 10 mM HEPES, 12 mM glucose, 0.5 mM CaCl$_2$, and 0.5 mM MgCl$_2$) and maintained in the same buffer. Following the addition of the luciferase substrate coelenterazine h (Biosynth), cells were stimulated with agonist. For the Venus-kras + Nluc BRET, 48 h post transfection, cells were washed once with HBSS buffer (GE Healthcare) and maintained in the same buffer. Prior to reading, Coelenterazine h (Biosynth) was added to a final concentration of 5 μM. BRET was read using a CLARIOstar (BMG LABTECH) microplate reader equipped with two monochromators to measure acceptor (535 ± 30 nm) and donor (475 ± 30 nm) emission signals. The BRET signal was determined as the ratio of light emitted by Venus-tagged biosensors (energy acceptors) and light emitted by Rluc8/Nluc-tagged biosensors (energy donors). Net BRET was calculated as the difference in BRET ratio from cells expressing donor alone with cells expressing both donor and acceptor. Venus-kras fluorescence was measured using a CLARIOstar microplate reader (excitation 497 ± 15 nm, emission 540 ± 20 nm) and calculated as average fluorescence from each control well.

**Flow cytometry.** HEK293 cells transiently transfected with SNAP-tagged FZD constructs and mG constructs or stably expressing SNAP-FZD$_6$ and SNAP-FZD$_6$ R416A$^{6.32}$ were grown in a 6-well plate. On the day of the experiment, the cells were detached with ice-cold 10 mM EDTA/PBS and then centrifuged at 400× g for 5 min in complete DMEM medium. The cells were resuspended in ice-cold 0.5% BSA/PBS, counted and transferred (3 × 10$^5$ cells) to a round-bottom 96-well plate. The plate was then centrifuged at 400× g for 5 min and subsequently cells were incubated with SNAP-substrate: either SNAP-Surface Alexa Fluor 488 (NEB #S9129S), SNAP-Surface Alexa Fluor 647 (NEB #S9136S) or SNAP-Cell 647-SiR (NEB #S9102S) at 1:200 dilution in complete DMEM medium for 30 min at 37 °C. The plate was centrifuged twice, cells were resuspended in ice-cold 0.5% BSA/PBS, and assayed immediately on an ADP Cyan flow cytometer. The median fluorescence intensity (MFI) data were analyzed using FlowJo V10 (Tree Star).

**Immunoblotting.** HEK293 cells were plated in 12 or 24-well plates. After 24 h, cells were transfected using Lipofectamine 2000 according to the manufacturer's instructions. Protein lysates were obtained using urea lysis buffer (0.5% NP-40, 2% SDS, 75 mM NaCl, 88 mM Tris/HCl, 4.5 M urea, 10% β-mercaptoethanol, 10% glycerol, pH 7.4). Lysates were sonicated and analyzed by 7.5, 10, or 4–20 % Mini-PROTEAN TGX precast polyacrylamide gels (Bio-Rad) and transferred to PVDF membranes using the Trans-Blot Turbo system (Bio-Rad). After blocking with 5% milk in TBS-T, membranes were incubated with primary antibodies in blocking buffer: rabbit anti-GAPDH (1:8000; Cell Signaling Technology #2118), rabbit anti-DVL2 (1:1000; Cell Signalling Technology #3216), rabbit-anti-P-S648 FZD$_6$ antibody (1:500; custom made), and rabbit anti-SNAP tag (1:1000, New England Biolabs #P9310S) overnight at 4 °C. The anti-P-S648 antibody was raised on a service basis by Moravian Biotechnology and validated previously[52]. Proteins were detected with horseradish peroxidase-conjugated secondary antibody (1:10000; goat anti-rabbit (Thermo Fisher Scientific #31460)) and Clarity Western ECL Blotting Substrate (Bio-Rad). All uncropped immunoblots can be found in the Supplementary Figure 11.

**AlphaScreen quantification of ERK1/2 and P-ERK1/2 levels.** Cells were seeded into a transparent 96-well plate at a density of 5 × 10$^4$ cells/well and allowed to adhere for over 6 h. The medium was then replaced and the cells were incubated with different concentrations of C59 or vehicle (DMSO) in serum-free DMEM at 37 °C overnight. The levels of ERK1/2 and P-ERK1/2 were assessed using the Alpha SureFire Ultra Multiplex assay kit (PerkinElmer) according to the manufacturer's instructions. Briefly, cells were lysed in 100 μl of SureFire Ultra lysis buffer and 10 μl of this lysate were added to wells of a 384-well light gray AlphaPlate (PerkinElmer). Subsequently, 5 μl of a mixture of SureFire Ultra reaction buffers 1 and 2, SureFire Ultra activation buffer and AlphaScreen acceptor beads were added to the lysate. Plates were incubated for 2 h at RT in the dark before the addition of 5 μl of suspension of donor beads in dilution buffer. The plate was then incubated for additional 2 h at RT in the dark before the luminescence signal was measured on an EnVision plate reader (PerkinElmer) using AlphaScreen mode with 535 and 615 nm emission filters.

**TOPFlash luciferase assay.** HEK293ΔFZD$_{1-10}$ cells[55] were seeded onto 48-well plates and the next day cells were transfected with M50 Super 8x TOPFlash (Addgene #12456), pRL-TK Luc (Promega E2241), FZD$_5$ and empty vector. 4 h post transfection, medium was changed to starvation medium with or without WNT-3A (300 ng ml$^{-1}$; Biotechne 5036-WN). 24 h after transfection, cells were analyzed by the Dual-Luciferase Reporter Assay System (Promega E1910)

according to manufacturer's instructions in white 96-well plates with the following modifications: cells were lysed in 50 μl Passive Lysis Buffer, Stop & Glo reagent was used at 0.5X and 25 μl of LARII and Stop & Glo Reagent were used for each well. Luminescence was measured using a Synergy2 microplate reader (BioTek).

**Live cell imaging**. HEK293 cells were seeded on 35 mm ECM gel-coated (1:300, Sigma-Aldrich) glass bottom dishes (Greiner Bio One 4 compartment 35 mm glass bottom dishes) at a density of $10^5$ cells/well. After 24 h, cells were transiently transfected using Lipofectamine 2000 according to the manufacturer's instructions with DVL2-GFP and either SNAP-FZD$_6$ or SNAP-FZD$_6$ R416A[6,32]. About 24 h post transfection, medium was removed and cells were incubated with SNAP-Cell 647-SiR (1:500) in BE buffer for 15 min, subsequently washed twice and imaged using a Zeiss LSM 710 confocal microscope.

**Relative variation score**. Sitewise relative variation score at each position $i$ was calculated as:

$$\mathrm{RelVar}_i = \log \frac{\frac{CV_i}{\max_j CV_j} + 1}{\frac{NV_i}{\max_j NV_j} + 1} \tag{1}$$

where $CV_i$ is the number of cancer variants at position $i$, $\max_j CV_j$ is the maximum number of cancer mutations at any position, and $NV_i$, is the number of naturally occurring variants at position $i$ and $\max_j NV_j$ is the maximum number of naturally occurring variants at any position.

**Phylogenetic analysis**. Phylogenetic tree for human FZD$_{1-10}$ and SMO was obtained by first aligning protein-coding sequences with the MAFFT aligner ran in the G-INS-i mode[82] and then performing phylogeny reconstruction in RAxML using the PROTGAMMALG substitution model[83].

**Multiple sequence alignment of Class F homologs**. Sequences for one-to-one orthologs for each Class F receptor in human were downloaded from Ensembl[84] for all species with the exception of *S. cerevisiae* using the BiomaRt package[85]. Orthologs with homology confidence 1 were retained and corresponding sequences were aligned using MAFFT in the G-INS-i mode (Supplementary Figure 1b and Supplementary Information).

**Statistical analysis**. Statistical and graphical analysis were performed using Graph Pad Prism software. Data were analyzed by two-tailed $t$-test or one-way ANOVA with Fisher's least significant difference (LSD) post-hoc analysis. Concentration-response curves of BRET data were fit using three, four parameter or bell-shaped non-linear regression. Significance levels are given as: $*P < 0.05$; $**P < 0.01$; $***P < 0.001$; $****P < 0.0001$. Data points throughout the manuscript are indicated as either the mean ± standard error of the mean (s.e.m.) or the mean ± standard deviation (s.d.).

**Reporting summary**. Further information on experimental design is available in the Nature Research Reporting Summary linked to this article.

## Data availability

Data supporting the findings of this manuscript are available from the corresponding author upon reasonable request. A reporting summary for this Article is available as a Supplementary Information file.

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

## Acknowledgements

We thank Anna Krook for access to the ClarioStar plate reader, Benoit Vanhollebeke for providing the HEK293TΔFZD$_{1-10}$ cells, and Roger Sunahara for permission to use and modify his receptor and heterotrimer cartoons. The work was supported by grants from Karolinska Institutet, the Swedish Research Council (2013-5708, 2015-02899, 2017-04676), the Swedish Cancer Society (CAN2014/659, CAN2017/561), the Novo Nordisk Foundation (NNF17OC0026940), the Knut & Alice Wallenberg Foundation (KAW2008.0149), Stiftelsen Olle Engkvist Byggmästare (2016/193), Emil and Wera Cornells Stiftelse, the Swedish Society for Medical Research (SSMF), the Science for Life Laboratory, the Marie Curie ITN WntsApp (Grant no. 608180; www.wntsapp.eu) and the Deutsche Forschungsgemeinschaft (DFG, German Research Foundation; grant no. KO 5463/1-1). M.M.S. received support from a FEBS Long-Term Fellowship. Computational resources were provided by the Swedish National Infrastructure for Computing (SNIC), High Performance Computing Centre North (HPC2N) in Umeå, National Supercomputer Centre (NSC) in Linköping, Uppsala Multidisciplinary Center for Advanced Computational Science (UPPMAX), and the Medical Research Council, UK (MC_U105185859).

## Author contributions

S.C.W., P.K., C.-F.B., M.K.-J. and N.O. performed wet lab experiments presented in the report. P.K., M.M.-S., D.R. and J.C. generated receptor models, structural alignments, and molecular dynamics simulation data. M.K.-J., J.P., S.C.W., P.K., C.-F.B., K.S., B.H. and J.V. prepared receptor constructs, mutants and performed validation of cellular receptor expression and function. Database analysis was done by Greg.S., M.M.S., Gunnar.S., M.M.B.. S.C.W., P.K., M.M.-S., Greg.S and Gunnar.S. prepared figures for publication. N.A.L. provided tagged mG proteins, FZD$_4$-NLuc, SMO-*R*luc8, Venus-kras, and expertize for setting up receptor-mG and bystander BRET. Gunnar.S., S.C.W., P.K. and M.M.B. wrote the manuscript with input from M.M.S., Greg.S., J.C., N.A.L.. Gunnar.S. supervised and coordinated the project.

## Additional information

**Competing interests:** The authors declare no competing interests.

