## [Peer Review File · Nature Communications]

Reviewers' Comments:

Reviewer #1:

Remarks to the Author:

This is an interesting manuscript that uses genomics information and molecular dynamics to identify a common mutation R6.32 found in the Class F family of GPCRs (FZD4, 5, 6, 7 and SMO), that is found in many human cancers. Furthermore, they investigate this mutation in FZD6 to explore whether it can convey constitutive activity and functional selectivity.

The evidence for constitutive receptor activity looks marginal to me and the authors did not use inverse agonists to try and confirm the constitutive nature of the activity. They did attempt to interfere with the local availability of native agonists. However, I wonder whether an alternative approach is to look at the impact of different levels of cell surface expression (perhaps using an inducible promoter) on receptor activity in the presence of inhibitors of agonist formation. A comparison of wild-type and mutant receptor might be very informative particularly if the emphasis of the manuscript is changed as suggested below.

What was much clearer from the experimental data presented was the negative effect of the mutation on coupling to signalling through DVL. In addition, strong evidence was provided that this mutation produced a clear increase in potency of agonists to receptor coupling to G protein mini genes. What is also clear is that in the wild-type receptors there is very poor coupling to the mini-G-protein alpha surrogates. Indeed, the conclusion I came to was that these receptors probably do not normally couple to heterotrimeric G proteins.

The emphasis of the manuscript is on a conserved activation mechanism for Class F receptors. I can understand this given the extent of the molecular dynamics and modelling presented in the manuscript. However, the main message for me is that this particular mutation (commonly found in many tumours) switches the normal coupling of this class of receptors from DVL to G-protein-coupled signalling. I therefore recommend that the emphasis of the manuscript is changed since I think this will be of wider interest to the readership of this journal. I would also change the title to emphasize this.

Reviewer #2:

Remarks to the Author:

The manuscript by Schulte and co-workers provides interesting insights into the mode of action of class F GPCRs (which includes FZDs and Smoothed), identifying the R/K6.32 amino acid in TM6 as the residue likely responsible for the switch in the TM6-TM7 interface, mediating activation of this class of GPCRs in their interaction with G protein transducers and resulting signaling pathways.

The manuscript starts with the intriguing identification of R/K6.32 in TM6 (and its partner W7.55 in TM7) as the mutational hot spot in cancers, and, in contrast, as a low natural variation residue. This analysis highlights the R/K6.32 - W7.55 pair as a potential molecular switch in the activation mechanism of class F GPCRs - in agreement with the recent structural work of Huang et al. (Cell 2018) on the activation mechanism of Smoothed. Structural modeling and dynamic simulation performed by Schulte and co-workers in the current manuscript further elaborates on this mechanism, suggesting that breakage of the R/K6.32 - W7.55 contact mediates opening of the receptors for downstream activation.

In order to test these predictions, and to glimpse into the potential mechanism of the cancer mutations in R/K6.32, the authors mutated this amino acid in FZD6 to Ala (and further - in other members of the family), expecting this R416A mutation to confer constitutive activity to FZD6. They performed several assays to monitor such activity, arguing for the constitutive nature of G protein engagement by this mutant receptor. In contrast, they found the mutant to be unable to engage Dvl - another immediate transducer of FZDs. The authors argue then that the structural

rearrangement within the R/K6.32 - W7.55 is key to receptor activation and selectivity in terms of downstream pathway activation, suggesting that these insights could be important for understanding of the oncogenic mechanisms in certain cancers, as well as subsequent drug discovery.

While definitely interesting and intriguing, this manuscript leaves many important issues not fully addressed. I list them below.

1. It is perhaps unfortunate that another structural work (in addition to Huang et al., Cell 2018 on Smoothed) has just appeared (Crystal structure of the Frizzled 4 receptor in a ligand-free state, Yang et al., Nature 2018). I am sure the authors have already verified their model with this new crystal structure. These considerations, along with the eventual comparisons of FZD4 with the other members of class F (keeping in mind the fact that FZD4, together with FZD9, have K in the R/K6.32 position, which is not mutated in cancers), should be added to the manuscript.

2. I find the data on the constitutive activity of FZD6[R416A] in the two functional assays, which should be key to the manuscript, not convincing enough:

2a. The first assay (Fig. 3a) measures Ca²⁺ mobilization. No traces of the Ca²⁺ signal are shown – they must be clearly added. Instead, only the bars are provided. It is not clear what they display – the height of the peak? The area under the peak? FZD6 transfection provides a modest 20% increase in the Ca²⁺ response upon application of Wnt5a, while transfection with FZD6[R416A] – a higher ca. 50% increase. Despite equal expression levels of the two FZD6 constructs, these findings are difficult to interpret, and they do not suggest a constitutive activity of FZD6[R416A], as the activity measured is the response to Wnt5a.

2b. The other assay is the monitoring of the ERK1/2 phosphorylation in HEK293 cells transfected with FZD6 or FZD6[R416A]. Instead of the conventional Western blots analyzing pERK levels, the authors chose the Perkin Elmer AlphaScreen assay – perhaps of its more quantitative nature. However, representative Westerns should still be added to this analysis. Using the AlphaScreen assay, the authors show that FZD6 transfection increases ca. 2-fold the levels of pERK, while FZD6[R416A] increases these levels ca. 3-fold. In order to exclude any potential autocrine Wnt production and FZD6 stimulation, the authors applied a Porcupine inhibitor C59, which reduced both FZD6- and FZD6[R416A]-induced increases in pERK. However, these levels still do not go down to the basal levels found in the cells not transfected with any FZD6 construct. The authors seem to argue that as the pERK levels in the C59-treated FZD6-expressing cells are not statistically significantly different from the basal levels, while the pERK levels in the C59-treated FZD6[R416A]-expressing cells are, this signifies the constitutive activity of FZD6[R416A]. This is wrong, it is simply a trick of wrongly interpreted statistics. It could have been equally wrongly argued that as there is no statistical significance in the difference between C59 treatment and control treatment of the FZD6-expressing cells, it must be concluded that FZD6 does not depend on any autocrine Wnt production and is, as such, constitutively active – to the opposite to what the authors wish to argue. In reality, the differences in the pERK levels in all these conditions are not strong enough to be able to draw any conclusions.

What also is left unaddressed is whether this FZD6 and FZD6[R416A]-induced pERK levels are sensitive to Gi inhibition with pertussis toxin. This is a simple missing control.

3. I find very convincing the body of the data describing the increased coupling of the R/K6.32-Ala mutants of class F GPCRs to their cognate mG protein sensors. It is intriguing that the mutants cease to interact with Dvl. The authors are right to argue that the breakage of the R/K6.32 - W7.55 link thus suggest the mechanism of switching from the FZD interaction with Dvl to the interaction with G proteins. However, the implications of this important point are not properly addressed. One open question is: if the R/K6.32-Ala mutation indeed stops to interact with Dvl, how come that these mutations are oncogenic, as the authors suggested in the beginning of their work? Dvl is an indispensable component of the canonical Wnt pathway, and overactivation of this pathway is carcinogenic in many tissues. In this regard, mutations in R/K6.32 should have been,

to the contrary, tumor-suppressive? Is it possible that these mutations are found in the metastatic tumors, and as such would reflect the overactivation of the non-canonical, and suppression of the canonical, Wnt signaling. If so, the breakage of the R/K6.32 - W7.55 link would signify not only – and not quite – the mechanism of activation of class F receptors, but rather, speaking about FZDs, the mechanism of switching these receptors from the canonical path to the non-canonical. This is a rather obvious hypothesis, not at all discussed by the authors. And this hypothesis has its further consequences. Namely – shouldn't the authors have measured the canonical readouts in their HEK293 cells transfected with FZD6 and its mutant version? Would they not expect FZD6 to induce e.g. beta-catenin stabilization, nuclear translocation, TopFlash activation? While the FZD6[R416A] version would not be expected to induce these? How about the migratory capacities of cells expressing FZD6[R416A] (or respective mutants in other FZDs)? Are they not expected to be stimulated, as comparison to the expression of the non-mutated FZD versions? Perhaps other cell lines, cancerous, would be needed to be used for such experiments.

In sum, I find this work preliminary. It is definitely interesting, but it will become much more solid and far-reaching in its importance if the authors go ahead along the lines mentioned above (and provide some more solid experimental data for some of their conclusions).

Reviewer #3:

Remarks to the Author:

Wright et al. study class F receptors combining genomic analysis, structure modeling, molecular dynamics simulations and resonance energy transfer-based experiments. Remarkably, authors detect a position at the lower part of TM6 that is significantly mutated in diverse human cancers (R/K6.32). Focusing on FZD6, the most prevalent variant in Class F receptors is R6.32Q. The relevance of this position for receptor functioning was assessed by site directed in silico and in vitro mutation introducing an alanine across several class F receptors.

The value of this study is to link higher incidences of cancer with a point mutation in R6.32 in class F receptors and its effect on receptor functioning. Providing a coherent connection could give the paper the impact to be considered for publication in Nature Communications. However, while I appreciate the systematic study, I have some major concerns in this respect.

1. It has not been clear to me why authors used an alanine mutation (R6.32A). Such mutation is very different from the most prevalent variant (R6.32Q) to cancer in class F receptors. In fact, it is possible that an alanine in position 6.32 behaves completely different than a glutamine in terms of receptor activation and coupling to intracellular effector proteins. Authors should experimentally study cancer-relevant variants in addition to the tested alanine mutants.
2. Using structural modeling authors show that favorable polar interactions between TM6 and TM7 are lost upon alanine mutation. This structural finding merges with increased activation of the alanine mutant receptor observed under different experimental conditions. However, this is not necessarily true for the most prevalent variant (R6.32Q) which has a polar side chain with the potential of TM7 interaction. Authors need to address this in more detail.
3. According to the authors, their structural models reveal cation-pi interactions with the side chain of W7.55. As the authors may know, modeling cation-pi interactions using molecular mechanics is a challenge due to the lack of explicit polarization or charge transfer effects. For that, they should move to quantum mechanical methods or for proteins more likely semi-empirical ones.
4. The authors define a conserved network of interactions in TM6/TM7, which serves as a molecular switch required for the full activation of transducer-bound class F receptors. To support this statement, authors should not only mutate R6.32 but also counter residues such as the W7.55.

Response to the reviewers is marked in red.

Changes in form of changed text, added figures and new data are highlighted in **bold**.

General summary

We are thankful to all three reviewers for their constructive criticism and positive judgement. The comments were valuable to improve the manuscript. According to the main point of critique and editorial recommendations, we have focused on broadening the study to include the cancer mutations of the molecular switch in addition to the experimental R^{6.32}A mutant. Also, we elaborated more on the functional selectivity of the molecular switch providing an explanation for conformational selection of FZD-mediated DVL- and G protein-dependent signalling.

Please find our detailed response below. . In short, we have:

1. Implemented the functional selectivity of the molecular switch for G proteins over DVL in the title and the manuscript
2. Added more data focusing on the analysis of naturally occurring molecular switch cancer mutants of residues R^{6.32} and W^{7.55} (for FZD₆ and SMO).
3. Added the analysis of a FZD isoform (FZD₅), which is also known to signal through the WNT/β-catenin pathway. Both DVL recruitment and a transcriptional reporter assay (TOPFlash) were included.

Reviewers' comments:

Reviewer #1 (Remarks to the Author):

This an interesting manuscript that uses genomics information and molecular dynamics to identify a common mutation R6.32 found in the Class F family of GPCRs (FZD4, 5, 6, 7 and SMO), that is found in many human cancers. Furthermore, they investigate this mutation in FZD6 to explore whether it can convey constitutive activity and functional selectivity.

We thank the reviewer for the appreciative comment.

The evidence for constitutive receptor activity looks marginal to me and the authors did not use inverse agonists to try and confirm the constitutive nature of the activity. They did attempt to interfere with the local availability of native agonists.

It is important to know that there are no orthosterically acting small molecules available that target Frizzleds, neither agonists, inverse agonists nor antagonists. Even anti-CRD antibodies have not been shown in a pharmacological context to show negative efficacy at the receptor. Constitutive activity is generally defined by inverse agonists – as the reviewer states – but this is not possible in the case of FZDs due to the lack of pharmacological tools.

However, I wonder whether an alternative approach is to look at the impact of different levels of cell surface expression (perhaps using an inducible promoter) on receptor activity in the presence of inhibitors of agonist formation. A comparison of wild-type and mutant receptor might be very informative particularly if the emphasis of the manuscript is changed as suggested below.

While this is definitely an interesting point and a valuable suggestion, we have not further pursued this angle in the revision. We see the question of the constitutive activity of the molecular switch mutants as a minor point. We have addressed it with the tools that are available and an indication of

constitutive activity of both the wild type and – to a larger extent – the molecular switch mutant was detectable in the presence of the porcupine inhibitor C59. However, the reviewer is correct – as stated below – that the key findings are the enhanced binding of the mG protein conformational sensors and the lack of DVL “coupling”.

What was much clearer from the experimental data presented was the negative effect of the mutation on coupling to signalling through DVL. In addition, strong evidence was provided that this mutation produced a clear increase in potency of agonists to receptor coupling to G protein mini genes. What is also clear is that in the wild-type receptors there is very poor coupling to the mini-G-protein alpha surrogates. Indeed, the conclusion I came to was that these receptors probably do not normally couple to heterotrimeric G proteins.

We are very happy that the reviewer appreciates these findings and perceives our results as “strong evidence”. We have spent about 10 years deciphering WNT-induced, heterotrimeric G protein-mediated responses in different biological systems (such as microglia-like cell lines and primary microglia) as well as in overexpression systems (see also a paper in *Sci Signaling*, which will appear in the first Dec issue of the journal on FZD₅). For a summary of the literature, please see our recent review in *TiPS*, Schulte and Wright 2018¹. With this introduction, I would like to agree with the reviewer in the statement that several of the FZDs exhibit poor resonance energy transfer to the engineered mini G proteins even though we have observed G protein coupling/signaling in cells that endogenously express FZD/G proteins or are transfected with FZD or G proteins or both^{2, 3, 4, 5, 6, 7, 8, 9, 10, 11}. On the other hand, I tend to disagree on the last statement because it goes against previous findings in both packaging and more physiologically relevant cells. However, our findings suggest that the FZD conformation that interacts with heterotrimeric G proteins is apparently very different from the one that is binding DVL. Despite being able to probe for the G protein-bound conformation of the receptor with mG BRET, translating the efficacy of these data to the ability of FZD paralogues or FZDs in general to interact better or worse than other GPCRs cannot be determined in the current setup. In other words, because the receptor itself is luciferase-tagged and involved in the readout, we cannot relate poor BRET efficiency with an inability to couple to G proteins. Moreover, there might be other mechanisms at play that regulate or inhibit the ability of FZDs to signal through G proteins – an active area of future research in our lab.

The emphasis of the manuscript is on a conserved activation mechanism for Class F receptors. I can understand this given the extent of the molecular dynamics and modelling presented in the manuscript. However, the main message for me is that this particular mutation (commonly found in many tumours) switches the normal coupling of this class of receptors from DVL to G-protein-coupled signalling. I therefore recommend that the emphasis of the manuscript is changed since I think this will be of wider interest to the readership of this journal. I would also change the title to emphasis this.

Thanks for this comment! We have indeed **changed the title and the manuscript** to address this issue. We have pursued this further and provided several additional experiments:

- **DVL recruitment of FZD₆ wt and the experimental R416A^{6.32} as well as the naturally occurring cancer mutants R416Q^{6.32} and W493L^{7.55} using bystander BRET**
- **DVL electrophoretic mobility shift in combination with the experimental R416A^{6.32} as well as the naturally occurring cancer mutants R416Q^{6.32} and W493L^{7.55}**
- **Detection of P-S648 in FZD₆ wt, R416A^{6.32}, R416Q^{6.32} and W493L^{7.55} as a functional outcome of the mutation according to recent findings by our group¹²**

- **DVL recruitment with FZD₅ wt and R449A^{6.32} using bystander BRET**
- **TOPFlash with FZD₅ wt and the experimental mutant FZD₅ R449A^{6.32} supporting the idea that the mutation of the molecular switch shows functional selectivity to G protein over DVL.**

Reviewer #2 (Remarks to the Author):

The manuscript by Schulte and co-workers provides interesting insights into the mode of action of class F GPCRs (which includes FZDs and Smoothened), identifying the R/K6.32 amino acid in TM6 as the residue likely responsible for the switch in the TM6-TM7 interface, mediating activation of this class of GPCRs in their interaction with G protein transducers and resulting signaling pathways. The manuscript starts with the intriguing identification of R/K6.32 in TM6 (and its partner W7.55 in TM7) as the mutational hot spot in cancers, and, in contrast, as a low natural variation residue. This analysis highlights the R/K6.32 - W7.55 pair as a potential molecular switch in the activation mechanism of class F GPCRs - in agreement with the recent structural work of Huang et al. (Cell 2018) on the activation mechanism of Smoothened. Structural modeling and dynamic simulation performed by Schulte and co-workers in the current manuscript further elaborates on this mechanism, suggesting that breakage of the R/K6.32 - W7.55 contact mediates opening of the receptors for downstream activation.

In order to test these predictions, and to glimpse into the potential mechanism of the cancer mutations in R/K6.32, the authors mutated this amino acid in FZD6 to Ala (and further – in other members of the family), expecting this R416A mutation to confer constitutive activity to FZD6. They performed several assays to monitor such activity, arguing for the constitutive nature of G protein engagement by this mutant receptor. In contrast, they found the mutant to be unable to engage Dvl – another immediate transducer of FZDs. The authors argue then that the structural rearrangement within the R/K6.32 - W7.55 is key to receptor activation and selectivity in terms of downstream pathway activation, suggesting that these insights could be important for understanding of the oncogenic mechanisms in certain cancers, as well as subsequent drug discovery.

While definitely interesting and intriguing, this manuscript leaves many important issues not fully addressed. I list them below.

1. It is perhaps unfortunate that another structural work (in addition to Huang et al., Cell 2018 on Smoothened) has just appeared (Crystal structure of the Frizzled 4 receptor in a ligand-free state, Yang et al., Nature 2018). I am sure the authors have already verified their model with this new crystal structure. These considerations, along with the eventual comparisons of FZD4 with the other members of class F (keeping in mind the fact that FZD4, together with FZD9, have K in the R/K6.32 position, which is not mutated in cancers), should be added to the manuscript.

While the original manuscript already contained the information of the recent SMO structures, we have now **added the new structural information obtained from the FZD₄ structure** (an apo-GPCR in the absence of both ligand and G protein, i.e. the structure presents an inactive state structure). In the FZD₄ structure, an additional contact between K^{6.32} in the molecular switch and TM2 (K^{6.32}-E^{2.41} contact) can be detected; the information from the FZD₄ structure further supports the existence of the closed molecular switch in the inactive receptor. Also, we think it is important to point out that the recently reported full-length SMO structures¹³ (PDB 6D35 and 6D32) are – despite the authors'

claims – to be seen as inactive structures because of the absence of the C5 helix of a heterotrimeric G protein (or any other transducer molecule)^{14, 15}.

2. I find the data on the constitutive activity of FZD6[R416A] in the two functional assays, which should be key to the manuscript, not convincing enough:

2a. The first assay (Fig. 3a) measures Ca²⁺ mobilization. No traces of the Ca²⁺ signal are shown – they must be clearly added. Instead, only the bars are provided. It is not clear what they display – the height of the peak? The area under the peak? FZD6 transfection provides a modest 20% increase in the Ca²⁺ response upon application of Wnt5a, while transfection with FZD6[R416A] – a higher ca. 50% increase. Despite equal expression levels of the two FZD6 constructs, these findings are difficult to interpret, and they do not suggest a constitutive activity of FZD6[R416A], as the activity measured is the response to Wnt5a.

The reviewer is completely correct that the calcium data do not show that wt or the R416A^{6,32} mutant of FZD₆ are constitutively active. We have indeed been very cautious in this regard and have only argued that the mutant (compared to wt) mediates a larger agonist-induced calcium response. Constitutive activity can pharmacologically only be nailed down by inverse agonists – in fact, the discovery of constitutive activity led to the definition of inverse agonists and negative efficacy. However, no inverse agonists or other orthosteric small molecules exist for Frizzleds. To circumvent this limitation, we have used C59 to create agonist-free conditions discussed in Figure 3b.

We have reworded the section heading, the corresponding figure title and the wording in the paragraph to be clearer about the chain of thoughts coming from an enhanced agonist-induced effect in the R^{6,32} to A mutant to the detection of constitutive activity in the AlphaScreen ERK1/2 assay in the absence of ligands.

The calcium measurements are based on Fluo3-AM fluorescence values from a plate reader at a fixed time point post stimulation (as stated in the method description). Therefore, no calcium traces were shown.

2b. The other assay is the monitoring of the ERK1/2 phosphorylation in HEK293 cells transfected with FZD6 or FZD6[R416A]. Instead of the conventional Western blots analyzing pERK levels, the authors chose the Perkin Elmer AlphaScreen assay – perhaps of its more quantitative nature. However, representative Westerns should still be added to this analysis.

Indeed the reviewer is correct in the assumption that we used the plate-based assay to obtain a more quantitative readout. Given that the responses are small and induced by receptor overexpression in a basal, agonist-independent manner, extracting differences with C59 required more sensitive methodology. According to the personnel from PerkinElmer, this assay has never been used to assess differences in receptor-induced, agonist-independent P-ERK1/2 levels before. In our eyes, it does not add any more value to provide representative immunoblots which are semiquantitative, because we have employed a much more sensitive and quantitative methodology in form of the AlphaScreen, an assay that relies on a similar set of ERK1/2 and P-ERK1/2 antibodies.

Using the AlphaScreen assay, the authors show that FZD6 transfection increases ca. 2-fold the levels of pERK, while FZD6[R416A] increases these levels ca. 3-fold. In order to exclude any potential autocrine Wnt production and FZD6 stimulation, the authors applied a Porcupine inhibitor C59, which reduced both FZD6- and FZD6[R416A]-induced increases in pERK. However, these levels still do not go down to the basal levels found in the cells not transfected with any FZD6 construct. The

authors seem to argue that as the pERK levels in the C59-treated FZD6-expressing cells are not statistically significantly different from the basal levels, while the pERK levels in the C59-treated FZD6[R416A]-expressing cells are, this signifies the constitutive activity of FZD6[R416A]. This is wrong, it is simply a trick of wrongly interpreted statistics. It could have been equally wrongly argued that as there is no statistical significance in the difference between C59 treatment and control treatment of the FZD6-expressing cells, it must be concluded that FZD6 does not depend on any autocrine Wnt production and is, as such, constitutively active – to the opposite to what the authors wish to argue. In reality, the differences in the pERK levels in all these conditions are not strong enough to be able to draw any conclusions.

What also is left unaddressed is whether this FZD6 and FZD6[R416A]-induced pERK levels are sensitive to Gi inhibition with pertussis toxin. This is a simple missing control.

We are very thankful for this comment from this reviewer. **We have reworded section headings, the figure title and corresponding passage in the text to clarify this issue.** The reviewer is correct that the changes in the P-ERK1/2 signal are small, but it should be emphasized that without inverse agonist, we cannot address constitutive activity through normal means and this is why we made use of the current setup.

- a) The basal levels of P-ERK1/2 are at the detection level of the Alpha Screen assay (stimulation of the serum-starved cells with 1 % serum led to a dramatic increase – not shown).
- b) Constitutive activity of GPCRs has been discovered in the context of receptor overexpression. Given receptor dynamics, wt receptors can show constitutive activity to some degree, depending also on the level of overexpression. The fact that C59 does not reduce the P-ERK1/2 levels in FZD₆ wt-transfected cells can be interpreted as basal constitutive activity of the overexpressed wt receptor. However, based on numbers and statistical significance, we did not elaborate on this. What was statistically significant was the remaining effect of FZD₆ R416A^{6,32} in the presence of C59 as a clear-cut indication of the receptor's constitutive activity.

The most interesting comparison in this experiment is not the difference between basal and receptor-induced in the presence of C59, but rather the effect of C59 in the receptor-induced situation.

In order to clarify this more, we have reworded the section emphasizing the tendency for constitutive activity of the overexpressed wt FZD₆.

- c) We would like to oppose the reviewer in the statement that “we wish to argue” against the constitutive activity of the wt FZD₆. In fact, it would only be logical and confirmed by many previous studies on other GPCRs that wt receptors in general display a certain degree of constitutive activity. The question is often, however, if our assays are sensitive enough to detect this. It is clearer, however, that we can detect the constitutive activity of the molecular switch mutant receptor. Statistics allow us only to argue for the constitutive activity of the mutant receptor, whereas visual inspection of the graph clearly argues also for a constitutive activity of the wt FZD₆ as pointed out by the reviewer.
- d) Regarding the control with the Gi/o protein inhibitor PTX. In several different settings, we have shown that the FZD₆-induced responses are mediated by Gi and Gq proteins and thus we expect a FZD₆-mediated composite response including both PTX-sensitive Gi and the PTX-insensitive Gq proteins^{3,6,7}. Given the negative effect of PTX on basal P-ERK1/2 in many cell types and the limited dependence of the FZD₆-mediated response to ERK1/2 on PTX-sensitive Gi/o, we feel that this experiment is not a “simple control” experiment and we did not include it in the revision.

3. I find very convincing the body of the data describing the increased coupling of the R/K6.32-Ala mutants of class F GPCRs to their cognate mG protein sensors. It is intriguing that the mutants cease to interact with Dvl. The authors are right to argue that the breakage of the R/K6.32 - W7.55 link thus suggest the mechanism of switching from the FZD interaction with Dvl to the interaction with G proteins. However, the implications of this important point are not properly addressed. One open question is: if the R/K6.32-Ala mutation indeed stops to interact with Dvl, how come that these mutations are oncogenic, as the authors suggested in the beginning of their work? Dvl is an indispensable component of the canonical Wnt pathway, and overactivation of this pathway is carcinogenic in many tissues. In this regard, mutations in R/K6.32 should have been, to the contrary, tumor-suppressive? Is it possible that these mutations are found in the metastatic tumors, and as such would reflect the overactivation of the non-canonical, and suppression of the canonical, Wnt signaling. If so, the breakage of the R/K6.32 - W7.55 link would signify not only – and not quite – the mechanism of activation of class F receptors, but rather, speaking about FZDs, the mechanism of switching these receptors from the canonical path to the non-canonical. This is a rather obvious hypothesis, not at all discussed by the authors. And this hypothesis has its further consequences. Namely – shouldn't the authors have measured the canonical readouts in their HEK293 cells transfected with FZD6 and its mutant version? Would they not expect FZD6 to induce e.g. beta-catenin stabilization, nuclear translocation, TopFlash activation? While the FZD6[R416A] version would not be expected to induce these? How about the migratory capacities of cells expressing FZD6[R416A] (or respective mutants in other FZDs)? Are they not expected to be stimulated, as comparison to the expression of the non-mutated FZD versions? Perhaps other cell lines, cancerous, would be needed to be used for such experiments.

We are very happy that the reviewer finds this part very convincing since this indeed presents the key finding of the paper. Guided by this reviewer's constructive criticism, we have added several new experiments. We will address the highly relevant questions of this reviewer here:

- a) Why are these mutants oncogenic? We would like to argue with two crucial points:
 - a. Smoothed does not interact with DVL and does not feed into WNT/ β -catenin signalling. Nevertheless, the SMO A1 mutant, which results in the disruption of the molecular switch by an exchange of W^{7.55} to L is highly oncogenic^{16, 17}.
 - b. One important aspect in this is that not only WNT/ β -catenin signalling can be oncogenic. Heterotrimeric G proteins (and GPCRs) are strong drivers of cancer as elegantly and comprehensively analysed by Silvio Gutkind¹⁸. Thus, the oncogenic potential of the molecular switch mutant could arise from overactive signalling through heterotrimeric G proteins especially in a tumor setting, where high levels of WNTs can be present. This is further underlined by the fact that FZD₆ does not activate the β -catenin-dependent signalling branch.
- b) The reviewer is incorrect in the assumption that FZD₆ would induce WNT/ β -catenin signalling resulting in a TOPFlash signal. FZD₆ does not signal in a β -catenin-dependent manner¹⁹ and thus, the suggested TOPFlash experiments are not feasible for this paralogue. In order to provide an additional DVL-dependent readout for the different FZD₆ constructs (FZD₆ wt, R416A^{6.32}, R416Q^{6.32} and W493L^{7.55}), we have taken advantage of a recently developed phospho-sensitive P-Ser648-FZD₆ antibody as a measure of receptor function with regard to interaction with a functional CK1 ϵ /DVL complex¹². Phosphorylation of Ser648 is observed in the wild type FZD₆ construct, but dramatically reduced in the FZD₆ molecular switch mutants, which reflects the reduced ability to recruit DVL as well as casein kinase 1, which

phosphorylates Ser648. **Data with the anti-P-Ser648 antibody for FZD₆ wt, R416A^{6.32}, R416Q^{6.32} and W493L^{7.55} are included in the new Fig. 4d, e.**

- c) In order to meet the reviewer's constructive and very valid request to investigate the impact of the molecular switch mutations on ligand-induced WNT/ β -catenin signalling, we have used FZD₅²⁰. **We provide new data on FZD₅ wt and molecular switch mutant (FZD₅ R449A^{6.32}) for DVL recruitment bystander BRET and TOPFlash reporter assays (comparing the WNT-3A-induced response in FZD₁₋₁₀ knockout cells transfected with FZD₅ wt or FZD₅ R449A^{6.32}). This extensive addition is now shown in Fig. 4f and g.**
- d) The current manuscript sheds light on the activation mechanisms of Class F receptors in a unique and novel way using genomics, mutagenesis, computational methodology and new conformational sensors. We feel that analysis of the cell biological consequences of the mutants exceeds the scope of this work. In addition, the pathophysiological relevance of the molecular switch mutants in Class F has already been addressed in the SMO mutants (SMO A1), which have been identified as driver mutants of sporadic basal cell carcinoma^{16,17}. Inclusion of the W^{7.55} mutants in the revised manuscript for SMO (and FZD₆) provides therefore an additional link to tumor biology.
- e) We have **updated the table listing the Class F molecular switch-related cancer mutations (FZD₁₋₁₀, SMO) for R^{6.32} and W^{7.55} (see Supplementary Figure 3b).**

In sum, I find this work preliminary. It is definitely interesting, but it will become much more solid and far-reaching in its importance if the authors go ahead along the lines mentioned above (and provide some more solid experimental data for some of their conclusions).

Reviewer #3 (Remarks to the Author):

Wright et al. study class F receptors combining genomic analysis, structure modeling, molecular dynamics simulations and resonance energy transfer-based experiments. Remarkably, authors detect a position at the lower part of TM6 that is significantly mutated in diverse human cancers (R/K6.32). Focusing on FZD6, the most prevalent variant in Class F receptors is R6.32Q. The relevance of this position for receptor functioning was assessed by site directed in silico and in vitro mutation introducing an alanine across several class F receptors.

The value of this study is to link higher incidences of cancer with a point mutation in R6.32 in class F receptors and its effect on receptor functioning. Providing a coherent connection could give the paper the impact to be considered for publication in Nature Communications. However, while I appreciate the systematic study, I have some major concerns in this respect.

We thank the reviewer for the positive tone and appreciation of the novel concept.

1. It has not been clear to me why authors used an alanine mutation (R6.32A). Such mutation is very different from the most prevalent variant (R6.32Q) to cancer in class F receptors. In fact, it is possible that an alanine in position 6.32 behaves completely different than a glutamine in terms of receptor activation and coupling to intracellular effector proteins. Authors should experimentally study cancer-relevant variants in addition to the tested alanine mutants.

While the R to Q mutant is restricted to FZD₃ and FZD₆, the other Class F members show R to H, C, S mutations (see **updated table on molecular switch-related cancer mutants in Class F receptors - Supplementary Figure 3**). Originally, we chose the Ala mutation of the molecular switch for reasons of simplicity and a routine mutagenesis approach. We do however appreciate the comment and have decided to more carefully investigate the function of the naturally occurring cancer mutations in the revised manuscript. In order to address this aspect carefully:

- a) We included a snapshot of FZD₆ R416Q as an additional panel in Fig. 2c.
- b) We extended the FZD₆-DVL bystander BRET data to present the full comparison of wt, R416A^{6.32}, R416Q^{6.32}, W493L^{7.55} (Fig. 4a).
- c) We extended the previous FZD₆-DVL electrophoretic mobility shift data to present the full comparison of wt, R416A^{6.32}, R416Q^{6.32}, W493L^{7.55} (Fig. 4b,c).
- d) We added the functional analysis of FZD phosphorylation at the Ser648 (see our recent JBC paper¹²) for FZD₆ wt, R416A^{6.32}, R416Q^{6.32} and W493L^{7.55} (Fig. 4d,e).
- e) We added FZD₅-DVL bystander BRET data for FZD₅ wt, and FZD₅ R449A^{6.32} (Fig. 4f)
- f) We added TOPFlash data for FZD₅ in combination with WNT-3A stimulation to compare FZD₅ wt and FZD₅ R449A^{6.32} in the WNT/ β -catenin-dependent pathway (Fig. 4g).
- g) We added mini G protein BRET experiments with prevalent SMO mutations, namely the SMO R451H^{6.32} and SMO W539L^{6.32}. With regard to the reviewer's argumentation about the R to Q mutation above, we have chosen H instead of Q because these mutations appear in more paralogues of the Class F than the R to Q mutation (**Supplementary Figure 8a**).
- h) We added mG protein BRET experiments with WNT-5A concentration response using FZD₆ and the molecular switch cancer mutations in FZD₆: R416Q^{6.32} and W493L^{7.55} (**Supplementary Figure 8b**).
- i) We added MD simulations on SMO, SMO R451A^{6.32}, SMO R451H^{6.32} and SMO W539L^{6.32} including RMSD plots and snapshots (**Supplementary Figure 8c-f**) zooming into the molecular switch region to visualize the change in the different SMO wt and cancer mutants over simulation time.

Thus, we feel that we have – according to the reviewer's recommendations - broadened the argumentation about (i) different cancer-associated mutations in the molecular switch, (ii) the effect of molecular switch mutation on FZDs feeding into WNT/ β -catenin signalling and (iii) diverse SMO mutants: R455H and W539L (the W539L^{7.55} has previously been defined as a driver mutant in sporadic basal cell carcinoma^{16,17}) validating our findings with the bona fide cancer driver mutant of SMO.

2. Using structural modeling authors show that favorable polar interactions between TM6 and TM7 are lost upon alanine mutation. This structural finding merges with increased activation of the alanine mutant receptor observed under different experimental conditions. However, this is not necessarily true for the most prevalent variant (R6.32Q) which has a polar side chain with the potential of TM7 interaction. Authors need to address this in more detail.

See comments and additions listed under comment #1

The cation- π interaction cannot be formed with a glutamine due to the lack of a cation - Q has a polar, uncharged side chain. In addition, this side chain is shorter than Arg or Lys and cannot form the interaction with the backbone of TM7.

In addition to the substantial experimental validation of the R416Q mutant of FZD₆ provided with the revision, we include here (not in the manuscript) results of additional MD simulation, which indicate that the glutamine does not form long-lived interactions with TM7 (very similar to what we observed for R^{6.32}A and W^{7.55}L). In **Fig. 2c**, we have included a model of the FZD₆ R416Q^{6.32}.

The figure shows snapshots from MD simulations focusing on the residues Q^{6.32} (left) and W^{7.55} (right) at 0, 75, 150 ns of simulation time.

3. According to the authors, their structural models reveal cation-pi interactions with the side chain of W7.55. As the authors may know, modeling cation-pi interactions using molecular mechanics is a challenge due to the lack of explicit polarization or charge transfer effects. For that, they should move to quantum mechanical methods or for proteins more likely semi-empirical ones.

It is correct that *ab initio* calculations at DFT-B3LYP and CCSD levels of theory have successfully been used to study the properties (e.g. energy) of intramolecular and intermolecular (e.g. ligand-protein) cation- π interactions as elegantly presented elsewhere. One has to note however that we have not claimed to attempt to model and further characterize the cation- π interactions between R/K^{6.32}-W^{7.55} over simulation time. Comparisons of quantum mechanical methods and molecular mechanics have demonstrated that classical force fields do capture π -cation interactions²¹. The location of the minimum of the potential energy profile is well captured, but the classical force fields underestimate depth of the minimum somewhat. A more accurate representation of this interaction (e.g. by a polarizable force field) could be expected to further stabilize the interaction between R^{6.32} and W^{7.55}, further strengthening our hypothesis that this is an important interaction for receptor function.

4. The authors define a conserved network of interactions in TM6/TM7, which serves as a molecular switch required for the full activation of transducer-bound class F receptors. To support this statement, authors should not only mutate R6.32 but also counter residues such as the W7.55.

As Class F representatives, we have chosen FZD₆ W493L^{6.32} and SMO W535L^{7.55} of which the latter is a well-characterized driver mutation in basal-cell carcinoma^{16,17}. Earlier experimental evidence underlines that the inverse agonist for SMO used in the present study (cyclopamine-KAAD) has an inhibitory effect on this mutant¹⁷. Please refer to point #1 for a list of experiments that were added in revision for the W^{7.55}L mutants of SMO and FZD₆.

References:

1. Schulte G, Wright SC. Frizzleds as GPCRs - More Conventional Than We Thought! *Trends Pharmacol Sci* **39**, 828-842 (2018).

2. Strakova K, *et al.* The tyrosine Y250(2.39) in Frizzled 4 defines a conserved motif important for structural integrity of the receptor and recruitment of Disheveled. *Cell Signal* **38**, 85-96 (2017).
3. Petersen J, *et al.* Agonist-induced dimer dissociation as a macromolecular step in G protein-coupled receptor signaling. *Nature communications* **8**, 226 (2017).
4. Hot B, *et al.* FZD10-Galpa13 signalling axis points to a role of FZD10 in CNS angiogenesis. *Cell Signal* **32**, 93-103 (2017).
5. Arthofer E, *et al.* WNT Stimulation Dissociates a Frizzled 4 Inactive-State Complex with Galpha12/13. *Mol Pharmacol* **90**, 447-459 (2016).
6. Kilander MB, *et al.* Disheveled regulates precoupling of heterotrimeric G proteins to Frizzled 6. *FASEB J* **28**, 2293-2305 (2014).
7. Kilander MB, Dahlstrom J, Schulte G. Assessment of Frizzled 6 membrane mobility by FRAP supports G protein coupling and reveals WNT-Frizzled selectivity. *Cell Signal* **26**, 1943-1949 (2014).
8. Halleskog C, Schulte G. Pertussis toxin-sensitive heterotrimeric G(α i/o) proteins mediate WNT/beta-catenin and WNT/ERK1/2 signaling in mouse primary microglia stimulated with purified WNT-3A. *Cell Signal* **25**, 822-828 (2013).
9. Halleskog C, *et al.* Heterotrimeric G protein-dependent WNT-5A signaling to ERK1/2 mediates distinct aspects of microglia proinflammatory transformation. *J Neuroinflammation* **9**, 111 (2012).
10. Kilander MBC, Halleskog C, Schulte G. Purified WNTs differentially activate beta-catenin-dependent and -independent pathways in mouse microglia-like cells. *Acta Physiologica* **203**, 363-372 (2011).
11. Kilander MBC, Dijksterhuis JP, Ganji RS, Bryja V, Schulte G. WNT-5A stimulates the GDP/GTP exchange at pertussis toxin-sensitive heterotrimeric G proteins. *Cellular Signalling* **23**, 550-554 (2011).
12. Strakova K, *et al.* Dishevelled enables casein kinase 1-mediated phosphorylation of Frizzled 6 required for cell membrane localization. *J Biol Chem*, (2018).
13. Huang P, *et al.* Structural Basis of Smoothed Activation in Hedgehog Signaling. *Cell* **174**, 312-324 e316 (2018).
14. Latorraca NR, Venkatakrisnan AJ, Dror RO. GPCR Dynamics: Structures in Motion. *Chemical reviews* **117**, 139-155 (2017).

15. Audet M, Bouvier M. Restructuring G-protein-coupled receptor activation. *Cell* **151**, 14-23 (2012).
16. Xie J, *et al.* Activating Smoothed mutations in sporadic basal-cell carcinoma. *Nature* **391**, 90-92 (1998).
17. Taipale J, *et al.* Effects of oncogenic mutations in Smoothed and Patched can be reversed by cyclopamine. *Nature* **406**, 1005-1009 (2000).
18. O'Hayre M, *et al.* The emerging mutational landscape of G proteins and G-protein-coupled receptors in cancer. *Nat Rev Cancer* **13**, 412-424 (2013).
19. Golan T, Yaniv A, Bafico A, Liu G, Gazit A. The human Frizzled 6 (HFz6) acts as a negative regulator of the canonical Wnt. beta-catenin signaling cascade. *J Biol Chem* **279**, 14879-14888 (2004).
20. Tauriello DV, *et al.* Wnt/beta-catenin signaling requires interaction of the Dishevelled DEP domain and C terminus with a discontinuous motif in Frizzled. *Proc Natl Acad Sci U S A* **109**, E812-820 (2012).
21. Khan HM, Grauffel C, Broer R, MacKerell AD, Jr., Havenith RW, Reuter N. Improving the Force Field Description of Tyrosine-Choline Cation-pi Interactions: QM Investigation of Phenol-N(Me)₄(+) Interactions. *J Chem Theory Comput* **12**, 5585-5595 (2016).

Reviewers' Comments:

Reviewer #1:

Remarks to the Author:

I think the authors have gone a long way towards dealing with my questions and concerns. However, I am still rather underwhelmed by the data presented in Figure 3. Given the calcium response is an endpoint read at 20 sec - there is a real risk that the peak response might be missed for such a transient response. I do think they should do a proper time course of these calcium transients, and perhaps also a concentration-response relationship.

Reviewer #2:

Remarks to the Author:

I am overall satisfied with the revisions done by the authors. I am particularly impressed by the new experimental data on FZD5, oncogenic mutations (such as FZD6[R416Q]), and reciprocal mutations in the W residue (such as FZD6[W493L]). The change in the main message of the story (the idea of the switch between Dvl- and G protein-dependent signaling modes) the authors now provide is very appropriate.

Although I still find unsatisfactory the discussion on the oncogenicity of the FZD R6.32 mutations (which switch the receptors to the G protein-interacting status from the Dvl-interacting status, which is counter-intuitive, given the established role of the Dvl-mediated canonical FZD signaling in oncogenic transformation), as well as the Calcium measurements (which still can be done as kinetics and not just 20sec-point, see my comments to the prior version of the manuscript for more details), these issues become minor on the background of the other advances of the paper.

I leave it to the editor to decide whether more revisions are necessary to address these two points, or whether the manuscript can be accepted as such. Both scenarios are fine with me.

Reviewer #3:

Remarks to the Author:

The authors have addressed all my concerns and comments.

Rebuttal letter

Reviewers' comments:

Reviewer #1 (Remarks to the Author):

I think the authors have gone a long way towards dealing with my questions and concerns. However, I am still rather underwhelmed by the data presented in Figure 3. Given the calcium response is an endpoint read at 20 sec - there is a real risk that the peak response might be missed for such a transient response. I do think they should do a proper time course of these calcium transients, and perhaps also a concentration-response relationship.

Reviewer #2 (Remarks to the Author):

I am overall satisfied with the revisions done by the authors. I am particularly impressed by the new experimental data on FZD5, oncogenic mutations (such as FZD6[R416Q]), and reciprocal mutations in the W residue (such as FZD6[W493L]). The change in the main message of the story (the idea of the switch between Dvl- and G protein-dependent signaling modes) the authors now provide is very appropriate.

Although I still find unsatisfactory the discussion on the oncogenicity of the FZD R6.32 mutations (which switch the receptors to the G protien-interacting status from the Dvl-interacting status, which is counter-intuitive, given the established role of the Dvl-mediated canonical FZD signaling in oncogenic transformation), as well as the Calcium measurements (which still can be done as kinetics and not just 20sec-point, see my comments to the prior version of the manuscript for more details), these issues become minor on the background of the other advances of the paper.

I leave it to the editor to decide whether more revisions are necessary to address these two points, or whether the manuscript can be accepted as such. Both scenarios are fine with me.

Vladimir Katanaev

Reviewer #3 (Remarks to the Author):

The authors have addressed all my concerns and comments.

Authors' comments to the reviewers:

We are very happy that the reviewers were satisfied with the revised version of the manuscript and the added data. We want to address the two remaining comments from rev #1 and #2:

1. Calcium data in Fig 3A:
 - a. Our results identify R^{6.32} as a molecular switch that could – in accordance to the Class A ionic lock – convey ligand-independent constitutive activity. In order to highlight this argumentation and to avoid confusion, we have removed the calcium data – which was criticised by both reviewers – in the second revision. The calcium data were showing the enhanced agonist-induced response in the R^{6.32} mutant of FZD₆ compared to wt, but did not address constitutive activity at all. In the current version, the logical flow and the transition from structural identification of the molecular switch (Fig. 1-2) to experimental confirmation and characterization feels more straightforward and to the point. Figure 3A and corresponding sentences in results and materials & method sections were removed.
2. Oncogenicity of molecular switch mutants in Class F:
 - a. We have introduced a paragraph in the discussion section addressing the oncogenic potential of molecular switch mutants in Class F. First, we highlight the molecular switch mutants of SMO as oncogenic drivers according to previous publications^{1, 2}. Second, we make clear that the detection of molecular switch mutants in human tumors does not prove their activity as tumor drivers, but that we exploit this information solely for mechanistic purposes of receptor activation. Further studies are now required to identify if and how these mutations contribute to oncogenesis.

In addition, we have introduced minor changes to comply with the journal's requirements for data presentation and manuscript style:

1. Bar graphs are overlaid with dot plots
2. Required statistical information is provided in the manuscript text/figure legends.
3. The title was shortened to 15 words.
4. The subheadings were changed to max 60 characters including spaces.
5. The abstract was shortened to 150 words.
6. A more elaborate data availability statement was added.
7. The forms Editorial Policy Checklist, Manuscript Checklist and Reporting summary are filled in and attached to the submission.

References

1. Xie J, *et al.* Activating Smoothed mutations in sporadic basal-cell carcinoma. *Nature* **391**, 90-92 (1998).
2. Taipale J, *et al.* Effects of oncogenic mutations in Smoothed and Patched can be reversed by cyclopamine. *Nature* **406**, 1005-1009 (2000).